# Reprogramming of GM-CSF-dependent alveolar macrophages through GSK3 activity modulation

Israel Ríos[1], Cristina Herrero[1], Mónica Torres-Torresano[2], Baltasar López-Navarro[2], María Teresa Schiaffino[2], Francisco Díaz Crespo[3], Alicia Nieto-Valle[4], Rafael Samaniego[4], Yolanda Sierra-Palomares[5], Eduardo Oliver[5], Fernando Revuelta-Salgado[6], Ricardo García-Luján[6], Paloma Sánchez-Mateos[4,7], Rafael Delgado[8,9], Amaya Puig-Kröger[2]*[†], Angel L Corbí[1]*[†]

[1]Myeloid Cell Laboratory, Centro de Investigaciones Biológicas, CSIC, Madrid, Spain; [2]Laboratorio de Inmuno-Metabolismo e Inflamación, Instituto de Investigación Sanitaria Gregorio Marañón (IiSGM), Hospital General Universitario Gregorio Marañón, Madrid, Spain; [3]Servicio de Anatomía Patológica, Hospital General Universitario Gregorio Marañón, Madrid, Spain; [4]Unidad de Microscopía Confocal, Instituto de Investigación Sanitaria Gregorio Marañón (IiSGM), Hospital General Universitario Gregorio Marañón, Madrid, Spain; [5]Experimental Pharmacology and New Tragets in Cardiopulmonary Disorders, Centro de Investigaciones Biológicas, CSIC, Madrid, Spain; [6]Servicio de Neumología. Hospital Universitario de Octubre, Madrid, Spain; [7]Department of Immunology, Ophthalmology and ENT, Universidad Complutense School of Medicine, Madrid, Spain; [8]Instituto de Investigación Hospital Universitario de Octubre (imas12), Madrid, Spain; [9]Universidad Complutense School of Medicine, Madrid, Spain

*For correspondence:
amaya.puig@iisgm.com (AP-K);
acorbi@cib.csic.es (ALC)

[†]These authors contributed equally to this work

## eLife Assessment

This **important** study provides **compelling** data from in vitro models and patient-derived samples to demonstrate how modulation of GSK3 activity can reprogram macrophages, revealing potential therapeutic applications in inflammatory diseases such as severe COVID-19. The study stands out for its clear and systematic presentation, **convincing** experimental approach, and the relevance of its findings to the field of immunology.

**Abstract** Monocyte-derived macrophages recruited into inflamed tissues can acquire an array of functional states depending on the extracellular environment. Since the anti-inflammatory/pro-fibrotic macrophage profile is determined by MAFB, whose activity/protein levels are regulated by GSK3, we addressed the macrophage reprogramming potential of GSK3 modulation. GM-CSF-dependent (GM-MØ) and M-CSF-dependent monocyte-derived macrophages (M-MØ) exhibited distinct levels of inactive GSK3, and inhibiting GSK3 in GM-MØ led to the acquisition of transcriptional, phenotypic, and functional properties characteristic of M-MØ (enhanced expression of IL-10 and monocyte-recruiting factors, and higher efferocytosis). These reprogramming effects were also observed upon GSK3α/β knockdown and through GSK3 inhibition in ex vivo isolated human alveolar macrophages (AMØ). Notably, GSK3 downmodulation potentiated the transcriptional signature of interstitial macrophages (IMØ) while suppressing the AMØ-specific gene profile. Indeed, heightened levels of inactive GSK3 and MAFB-dependent proteins were observed in severe COVID-19 patients'

lung macrophages, highlighting the GSK3-MAFB axis as a therapeutic target for macrophage reprogramming.

## Introduction

Macrophages exhibit considerable functional plasticity during inflammatory responses, transitioning from pro-inflammatory activities to tissue repair and inflammation resolution (*Dick et al., 2022*; *Mulder et al., 2021*). This functional versatility of macrophages is intricately tied to factors such as their ontogeny (fetal origin vs. monocyte-derived) and the specific tissue and extracellular environment (*Hoeffel and Ginhoux, 2018*; *Ruffell and Coussens, 2015*; *Lavin et al., 2015*). Notably, monocyte-derived macrophages are oppositely instructed by M-CSF or GM-CSF (*Caescu et al., 2015*; *Van Overmeire et al., 2016*; *Fleetwood et al., 2007*; *Lacey et al., 2012*; *González-Domínguez et al., 2016*; *Fleetwood et al., 2009*; *Sierra-Filardi et al., 2011*). GM-CSF directs monocyte-derived macrophages (GM-MØ) toward heightened pro-inflammatory activity (*González-Domínguez et al., 2015*; *Soler Palacios et al., 2015*; *de las Casas-Engel et al., 2013*) and the acquisition of the lung alveolar macrophage phenotype and gene profile (*Nieto et al., 2018*). In fact, GM-CSF is essential for generating lung alveolar macrophages (*Guilliams et al., 2013*), crucial for surfactant homeostasis, whereas deficient GM-CSF signaling leads to pulmonary alveolar proteinosis (*Trapnell et al., 2009*), a condition marked by defective surfactant clearance and disruption of pulmonary homeostasis due to alveolar macrophage dysfunction (*Huang et al., 2023*). Conversely, M-CSF gives rise to anti-inflammatory, pro-resolving, and immunosuppressive monocyte-derived macrophages (M-MØ) (*Mulder et al., 2021*; *Donadon et al., 2020*) but promotes the development of pro-fibrotic pathogenic monocyte-derived macrophages in lung pathologies like idiopathic pulmonary fibrosis (*Joshi et al., 2020*) and severe COVID-19 (*Vega et al., 2020*). The pathological relevance of GM-MØ and M-MØ subsets is evident in severe COVID-19, with a huge increase in monocyte-derived M-MØ and a reduction of tissue-resident GM-MØ-like lung macrophages (*Wendisch et al., 2021*; *Liao et al., 2020*; *Grant et al., 2021*).

Despite their crucial role in maintaining tissue homeostasis, deregulated macrophage functional specialization contributes to various human diseases, including tumors and chronic inflammatory pathologies (*Schultze, 2016*). Consequently, macrophage reprogramming has emerged as a promising therapeutic approach (*Schultze, 2016*). However, achieving this necessitates a profound understanding of the transcriptional and signaling mechanisms governing the pro- and anti-inflammatory nature of macrophages. In this regard, recent findings indicate that the homeostatic and reparative transcriptional profile of human M-MØ is orchestrated by MAF and MAFB (*Cuevas et al., 2017*; *Kim, 2017*; *Kang et al., 2017*), closely related transcription factors that regulate stemness and self-renewal in the mouse hematopoietic lineage (*Eychène et al., 2008*; *Lavin et al., 2014*; *Kelly et al., 2000*; *Aziz et al., 2009*; *Soucie et al., 2016*; *Sarrazin et al., 2009*). In fact, the distinctive expression of MAFB in M-MØ, influencing IL-10 production (*Cuevas et al., 2017*; *Cuevas et al., 2022*), also mediates macrophage reprogramming induced by methotrexate (*Ríos et al., 2023*) and LXR ligands (*de la Aleja et al., 2023*).

Given the pivotal role of MAFB/MAF in human macrophage functional specification, and considering the regulation of large MAF factors through GSK3-mediated phosphorylation-induced proteasomal degradation (*Eychène et al., 2008*), we hypothesized that modulating GSK3 activity could offer a viable avenue for reprogramming monocyte-derived macrophages. Our findings reveal significant differences in inhibitory phosphorylation of GSK3β (Ser$^9$-GSK3β) and GSK3α (Ser$^{21}$-GSK3α) between M-MØ and GM-MØ, and that GSK3 inhibition (using CHIR-99021) or knockdown (using siRNA) in GM-MØ prompts the adoption of transcriptional, phenotypic, and functional characteristics resembling those of M-MØ. This reprogramming effect was manifested in elevated expression of IL-10, monocyte-recruiting chemokines, pro-fibrotic factors, as well as in enhanced phagocytic capability, all correlating with increased MAFB expression. The reprogramming action of GSK3 inhibition was also observed in control monocytes and in monocytes exposed to a pathological pro-inflammatory environment. Importantly, GSK3 inhibition in ex vivo isolated human alveolar macrophages (AMØ) led to the loss of the AMØ gene signature and the acquisition of the gene profile that characterizes interstitial macrophages (IMØ). Since pathogenic lung macrophages in severe COVID-19 exhibit high levels of inactive GSK3 and heightened levels of MAFB and MAFB-dependent proteins, our

results underscore the potent macrophage reprogramming impact of GSK3 inhibition and highlight the GSK3-MAFB axis as a promising therapeutic target for macrophage reprogramming.

## Results

### Monocyte-derived macrophages generated in the presence of GM-CSF (GM-MØ) or M-CSF (M-MØ) differ in the state of activation of GSK3α and GSK3β

Monocyte-derived M-MØ exhibit a transcriptional profile akin to that of pathogenic macrophages in severe COVID-19 (*Vega et al., 2020*), whereas GM-MØ resemble lung alveolar macrophages (AMØ) (*Vega et al., 2020*). Our prior research has established that MAFB determines the transcriptional and functional specification of monocyte-derived M-MØ, characterized by significantly higher MAFB levels compared to GM-MØ (*Cuevas et al., 2017*). Since the protein levels of large MAF family members (MAF, MAFA, MAFB, NRL) are controlled through GSK3 phosphorylation-dependent proteasome degradation (*Eychène et al., 2008*), we initially assessed the relative levels of the inhibitory (Ser$^{21}$ for GSK3α and Ser$^9$ for GSK3β) and activation-associated (Tyr$^{279}$ for GSK3α and Tyr$^{216}$ for GSK3β) phosphorylation of GSK3 in M-MØ and GM-MØ. Ser$^{21}$-GSK3α and Ser$^9$-GSK3β phosphorylation was more pronounced in M-MØ, aligning with their elevated levels of MAFB (*Figure 1A*). Conversely, no significant difference in Tyr$^{279}$-GSK3α and Tyr$^{216}$-GSK3β phosphorylation was observed between M-MØ and GM-MØ (*Figure 1B*). Therefore, the heightened expression of MAFB in M-MØ correlates with a higher inhibitory phosphorylation of GSK3.

### Inhibition of GSK3 leads to transcriptional, phenotypic, functional, and metabolic reprogramming in monocyte-derived GM-MØ

Building upon the above findings, we aimed to examine the macrophage reprogramming effects of GSK3 activity modulation in GM-MØ through the use of the GSK3 inhibitor CHIR-99021 (*Figure 1C*). Kinetics and dose-response analysis of the effects of CHIR-99021 on MAFB expression showed that maximal protein levels were achieved after a 24–48 hr exposure to 10 μM CHIR-99021 (*Figure 1—figure supplement 1*), conditions that were used hereafter. Indeed, treatment of GM-MØ with CHIR-99021 for 48 hr resulted in a significant decrease in the activation-associated Tyr$^{216}$-GSK3β phosphorylation (*Figure 1D*) and a robust elevation in MAFB protein levels (*Figure 1E*). Moreover, reversal of the Tyr$^{216}$/Ser$^9$ GSK3β phosphorylation ratio was also seen after CHIR-99021 treatment, albeit it was not observed after exposure to other GSK3 inhibitors (SB-216763 or LiCl; *Figure 1—figure supplement 2*). RNA-Seq analysis of CHIR-GM-MØ unveiled a huge transcriptional impact of GSK3 inhibition, with more than 2000 differentially expressed genes ([log$_2$FC]>1; adjp<0.05) (*Figure 1F*). Corresponding with the higher MAFB expression, gene set enrichment analysis (GSEA) showed that the CHIR-GM-MØ transcriptome was significantly enriched in genes that either mark M-MØ (GSE68061) or are upregulated during monocyte-to-M-MØ differentiation (*Figure 1G*). Conversely, GM-MØ-specific genes (GSE68061) or genes increased during monocyte-to-GM-MØ differentiation were downregulated (*Figure 1G*). Additional gene ontology analysis further supported the pathophysiological significance of the macrophage reprogramming induced by modulating GSK3 activity. Thus, analysis of the MoMac-VERSE (a resource that identified conserved monocyte and macrophage states derived from healthy and pathologic human tissues) (GSE178209) *Mulder et al., 2021* indicated that GSK3 inhibition augments the expression of the gene sets that define MoMac-VERSE subsets identified as long-term resident macrophages [Cluster HES1_Mac (#2)] and tumor-associated macrophages with an M2-like signature [Clusters HES1_Mac (#2), TREM2_Mac (#3), C1Q$^{hi}$_Mac (#16) and FTL_Mac (#17)] (*Mulder et al., 2021*; *Figure 1H*). Moreover, GSK3 inhibition increased the expression of the gene sets that define pathogenic pro-fibrotic macrophage subsets in severe COVID-19 ([Group 3 or SPP1$^+$, GSE145926] [*Liao et al., 2020*], MoAM3 [GSE155249] [*Grant et al., 2021*] or CD163$^+$/LGMN$^+$ MØ [EGAS00001005634] [*Wendisch et al., 2021*]) (*Figure 1I*). Therefore, GSK3 inhibition enhances MAFB expression and reprograms macrophages by promoting the acquisition of the gene profile that defines anti-inflammatory and pro-fibrotic macrophages.

In agreement with the increased MAFB levels detected in CHIR-GM-MØ (*Figure 1E*), screening for mediators of the reprogramming action of GSK3 inhibition using DoRoThEA (*Garcia-Alonso et al., 2019*; https://saezlab.github.io/dorothea/; *Badia-i-Mompel et al., 2019*) revealed a significant

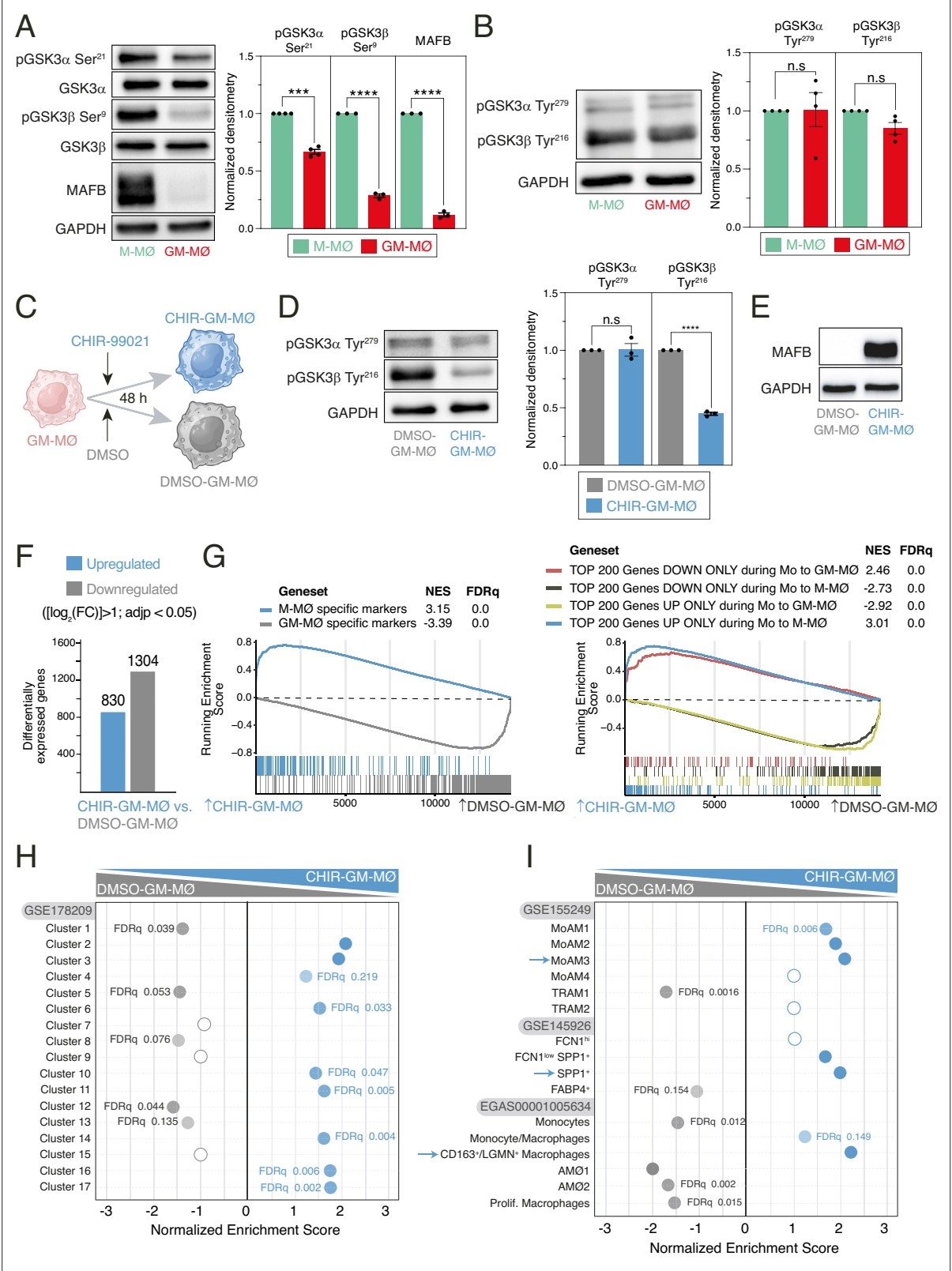

**Figure 1.** Transcriptional effect of GSK3 inhibition on monocyte-derived GM-MØ. (**A**) MAFB, total GSK3α, total GSK3β, Ser⁹-phosphorylated GSK3β, and Ser²¹-phosphorylated GSK3α levels in M-MØ and GM-MØ, as determined by western blot (left panel). GAPDH protein levels were determined as protein loading control. Mean ± SEM of the MAFB/GAPDH, Ser²¹-phosphorylated GSK3α/total GSK3α, and Ser⁹-phosphorylated GSK3β/total GSK3β protein ratios from three independent experiments are shown (right panel) (paired Student's t-test: ***, p<0.005; ****, p<0.001). A representative

*Figure 1 continued on next page*

*Figure 1 continued*

western blot experiment is shown in each case in the upper panel. (**B**) p-Tyr$^{279}$-GSK3α and p-Tyr$^{216}$-GSK3β levels in four independent samples of M-MØ and GM-MØ, as determined by western blot (left panel). GAPDH protein levels were determined as protein loading control. Mean ± SEM of the p-Tyr$^{279}$-GSK3α/GAPDH and p-Tyr$^{216}$-GSK3β/GAPDH protein ratios from four independent experiments are shown (right panel) (paired Student's t-test: n.s., not significant). (**C**) Schematic representation of the exposure of GM-MØ to CHIR-99021 for 48 hr. Figure 1C was created with BioRender. com. (**D**) Tyr$^{279}$-phosphorylated GSK3α and Tyr$^{216}$-phosphorylated GSK3β levels in three independent preparations of DMSO- or CHIR-99021-treated GM-MØ, as determined by western blot (left panel). GAPDH protein levels were determined as protein loading control. Mean ± SEM of the p-Tyr$^{216}$-GSK3β/GAPDH and p-Tyr$^{279}$-GSK3α/GAPDH protein ratios from the three independent experiments are shown (right panel) (paired Student's t-test: ****, p<0.001). (**E**) MAFB protein levels in DMSO-GM-MØ and CHIR-GM-MØ, as determined by western blot. A representative experiment is shown. (**F**) Number of differentially expressed genes ([log2FC]>1; adjp<0.05) between DMSO-GM-MØ and CHIR-GM-MØ. Differential gene expression was assessed using DESeq2. (**G**) Gene set enrichment analysis (GSEA) of the indicated gene sets (from GSE68061, left panel; from GSE188278, right panel) on the ranked comparison of the CHIR-GM-MØ vs. DMSO-GM-MØ transcriptomes. Normalized enrichment score (NES) and FDRq values are indicated in each case. (**H**) Summary of GSEA of the gene sets that define the tissue-resident monocyte and macrophage states (MoMac-VERSE) *Mulder et al., 2021* on the ranked comparison of the CHIR-GM-MØ vs. DMSO-GM-MØ transcriptomes. FDRq values are indicated only if FDRq>0.0; empty dots, not significant. (**I**) Summary of GSEA of the gene sets that characterize the macrophage subsets identified in severe COVID-19 (*Wendisch et al., 2021*; *Liao et al., 2020*; *Grant et al., 2021*) on the ranked comparison of the CHIR-GM-MØ vs. DMSO-GM-MØ transcriptomes. FDRq values are indicated only if FDRq>0.0; empty dots, not significant.

The online version of this article includes the following source data and figure supplement(s) for figure 1:

**Source data 1.** PDF file containing original western blots for *Figure 1A*, indicating the relevant bands.

**Source data 2.** Original files for western blot analysis displayed in *Figure 1A*.

**Source data 3.** Protein ratios of MAFB/GAPDH, p-Ser21-GSK3α/total GSK3α, and p-Ser9-GSK3β/total GSK3β in M-MØ and GM-MØ from three to five independent experiments and statistical analysis.

**Source data 4.** PDF file containing original western blots for *Figure 1B*, indicating the relevant bands.

**Source data 5.** Original files for western blot analysis displayed in *Figure 1B*.

**Source data 6.** Protein ratios of p-Tyr279-GSK3α/GAPDH and p-Tyr216-GSK3β/GAPDH in M-MØ and GM-MØ from four independent experiments and statistical analysis.

**Source data 7.** PDF file containing original western blots for *Figure 1D*, indicating the relevant bands.

**Source data 8.** Original files for western blot analysis displayed in *Figure 1D*.

**Source data 9.** Protein ratios of p-Tyr216-GSK3β/GAPDH and p-Tyr279-GSK3α/GAPDH in DMSO-GM-MØ and CHIR-GM-MØ from three independent experiments and statistical analysis.

**Source data 10.** PDF file containing original western blots for *Figure 1E*, indicating the relevant bands.

**Source data 11.** Original files for western blot analysis displayed in *Figure 1E*.

**Figure supplement 1.** MAFB protein levels in GM-MØ exposed to DMSO or distinct CHIR-99021 concentrations (0.1 µM, 1 µM, 10 µM) for 6–48 hr, as determined by western blot (left panel).

**Figure supplement 2.** p-Ser$^9$-GSK3β and p-Tyr$^{216}$-GSK3β levels in GM-MØ treated with DMSO, CHIR-99021 (10 µM), SB-216763 (10 µM) or LiCl (10 mM) for 48 hr, as determined by western blot.

enrichment in MAFB-regulated genes (*Figure 2A*), and similar results were yielded upon Enrichr analysis (*Kuleshov et al., 2016*; *Chen et al., 2013*) (https://maayanlab.cloud/Enrichr/) (*Figure 2—figure supplement 1*). In fact, GSEA confirmed that CHIR-GM-MØ are enriched in MAFB-dependent genes (*Vega et al., 2020*; *Cuevas et al., 2017*; *Figure 2B and C*) along with macrophage genes whose regulatory regions are directly bound by MAFB (*Simón-Fuentes et al., 2023*) (75-geneset, *Figure 2C*), including *CCL2*, *CCL8*, *CCL18*, *IGF1*, *IL10*, and *LGMN* (*Figure 2D*). Importantly, all these changes were reflected in the phenotype of CHIR-GM-MØ, which showed elevated expression of transcription factors like MAF (*Figure 2E*), cell surface proteins like CD163 and FOLR2 (*Figure 2E and F*), and soluble factors like CCL2, CCL18, IL-10, LGMN, and SPP1 (*Figure 2G*), all of which are either direct MAFB targets (*Simón-Fuentes et al., 2023*) or MAFB-regulated genes (*Vega et al., 2020*; *Cuevas et al., 2017*). Likewise, the attenuated expression of the GM-MØ-specific *INHBA* gene (*Figure 2D*) corresponded to a decrease in Activin A production by CHIR-GM-MØ (*Figure 2G*). These results underscore the phenotypic reprogramming induced by GSK3 inhibition in GM-MØ, wherein there is an upregulation of MAFB-dependent transcription factors, cell surface markers, and soluble factors which collectively define an anti-inflammatory/pro-fibrotic phenotype characteristic of M-CSF-dependent monocyte-derived macrophages.

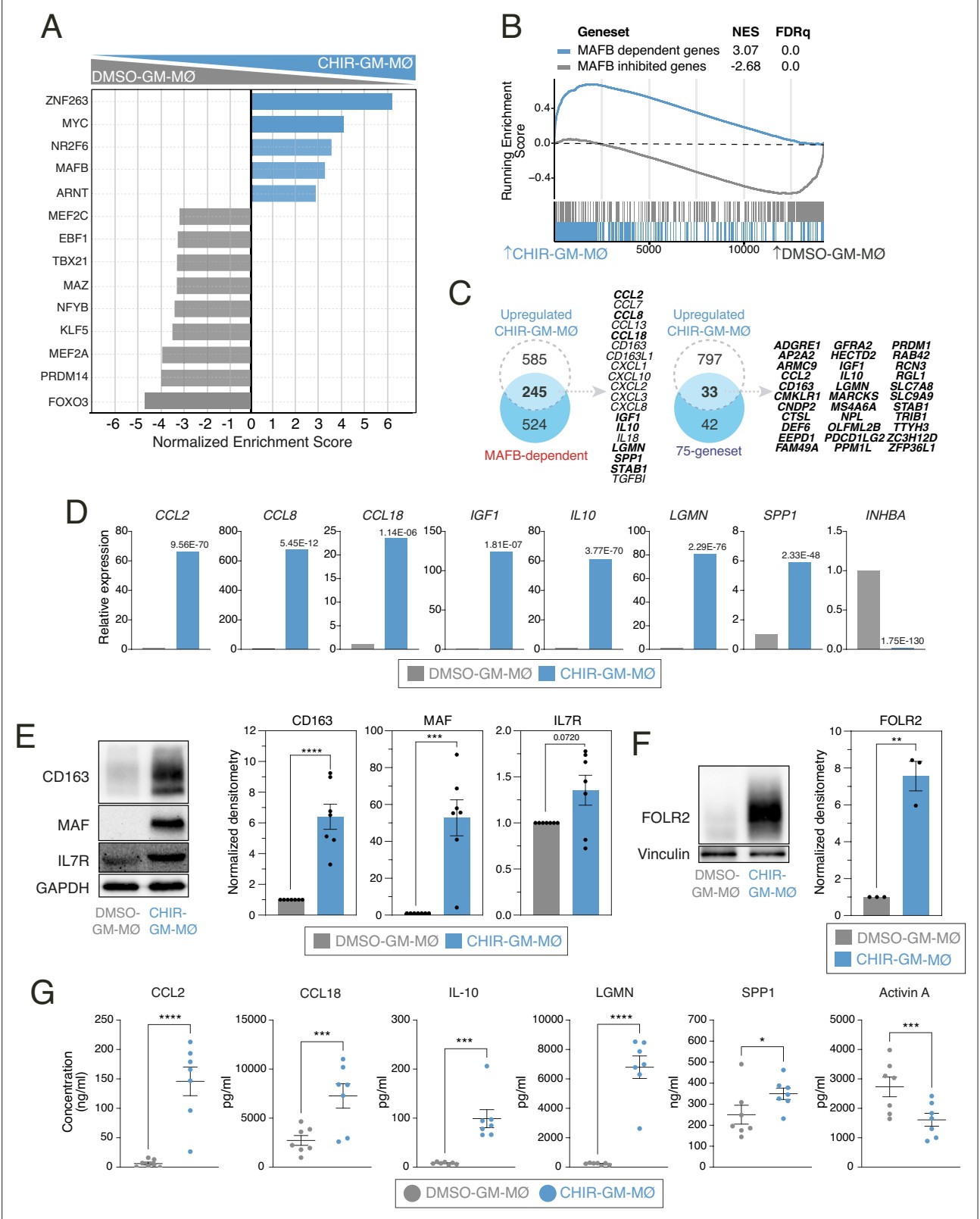

**Figure 2.** Phenotypic effects of GSK3 inhibition on monocyte-derived GM-MØ: enhanced expression of MAFB-dependent genes and proteins. (**A**) DoRoThEA (*Garcia-Alonso et al., 2019*) analysis on the ranked comparison of the DMSO-GM-MØ and CHIR-GM-MØ transcriptomes. (**B**) Gene set enrichment analysis (GSEA) of MAFB-regulated gene sets (from GSE155719) on the ranked comparison of the DMSO-GM-MØ and CHIR-GM-MØ transcriptomes. NES and FDRq values are indicated in each case. (**C**) Overlap between the genes upregulated (|log2FC|>1; adjp<0.05) in CHIR-GM-

*Figure 2 continued on next page*

*Figure 2 continued*

MØ (relative to DMSO M-MØ) and MAFB-dependent genes (from GSE155719, left panel) or the 75-geneset of MAFB-regulated genes (right panel) (*Simón-Fuentes et al., 2023*), with indication of some of the overlapping genes. (**D**) Relative mRNA levels of the indicated genes in DMSO-GM-MØ and CHIR-GM-MØ, as determined by RNA-Seq on three independent samples (GSE256208). Adjp of the comparison, shown in each case, was assessed using DESeq2. (**E**) CD163, MAF, and IL7R protein levels in DMSO-GM-MØ and CHIR-GM-MØ, as determined by western blot (left panel). GAPDH protein levels were determined as protein loading control. Mean ± SEM of the MAF/GAPDH, IL7R/GAPDH, and CD163/GAPDH protein ratios from seven independent experiments are shown (right panels) (paired Student's t-test: ***, p<0.005; ****, p<0.001). A representative western blot experiment is shown in each case. (**F**) FOLR2 protein levels in DMSO-GM-MØ and CHIR-GM-MØ, as determined by western blot (left panel). Vinculin protein levels were determined as protein loading control. Mean ± SEM of the FOLR2/Vinculin protein ratio from three independent experiments is shown (lower panel) (paired Student's t-test: **, p<0.01). A representative western blot experiment is shown. (**G**) Production of the indicated soluble factors by DMSO-GM-MØ and CHIR-GM-MØ, as determined by ELISA. Mean ± SEM of seven independent samples are shown (paired Student's t-test: *, p<0.05; **, p<0.01; ***, p<0.005; ****, p<0.001).

The online version of this article includes the following source data and figure supplement(s) for figure 2:

**Source data 1.** Discriminant regulon expression analysis (DoRothEA) of DMSO-GM-MØ and CHIR-GM-MØ transcriptomes.

**Source data 2.** mRNA levels (read counts) of the indicated genes in DMSO-GM-MØ and CHIR-GM-MØ, as determined by RNA-Seq on three independent experiments.

**Source data 3.** PDF file containing original western blots for *Figure 2E*, indicating the relevant bands.

**Source data 4.** Original files for western blot analysis displayed in *Figure 2E*.

**Source data 5.** Protein ratios of MAF/GAPDH, IL7R/GAPDH, and CD163/GAPDH in DMSO-GM-MØ and CHIR-GM-MØ from seven independent experiments and statistical analysis.

**Source data 6.** PDF file containing original western blots for *Figure 2F*, indicating the relevant bands.

**Source data 7.** Original files for western blot analysis displayed in *Figure 2F*.

**Source data 8.** Protein ratios of FOLR2/Vinculin in DMSO-GM-MØ and CHIR-GM-MØ from three independent experiments and statistical analysis.

**Source data 9.** Concentration of CCL2, CCL18, IL10, LGMN, SPP1, and Activin A in DMSO-GM-MØ and CHIR-GM-MØ from seven independent experiments and statistical analysis.

**Figure supplement 1.** Gene ontology analysis of the genes upregulated (|log2FC|>1; adjp<0.05) in CHIR-GM-MØ (relative to DMSO GM-MØ) using Enrichr and the indicated databases.

Remarkably, the phenotypic changes induced by GSK3 inhibition were also accompanied by alterations in macrophage functional capabilities. Thus, CHIR-GM-MØ exhibited elevated anti-inflammatory activity, evidenced by increased production of IL-10 upon activation by inflammatory cytokines such as TNF or IFNγ, or exposure to PAMPs like LPS or the TLR7 ligand CL264 (*Figure 3A*). In addition, CHIR-GM-MØ demonstrated increased phagocytic ability for *Escherichia coli* particles (*Figure 3B*), as well as enhanced efferocytosis (*Figure 3C*), with both activities being typically higher in M-MØ. Collectively, these findings indicate that GSK3 inhibition re-educates GM-MØ macrophages by facilitating the acquisition of transcriptional, phenotypic, and functional properties characteristic of M-CSF-dependent monocyte-derived macrophages. Of note, in line with the well-established connection between inflammatory potential and metabolic state in macrophages (*Wculek et al., 2022*), GSK3 inhibition also induced modifications in various metabolic parameters in GM-MØ, including basal respiration and maximal ATP production (*Figure 3—figure supplements 1–3*).

## siRNA-mediated knockdown of *GSK3A* and/or *GSK3B* enhances the expression of the MAFB-dependent profiles of M-MØ and pathogenic pro-fibrotic lung macrophages

To fully support the involvement of GSK3 in the macrophage reprogramming action of CHIR-99021, we next assessed the transcriptional consequences of knocking down *GSK3A* (coding for GSK3α) and/or *GSK3B* (coding for GSK3β) in GM-MØ (*Figure 4A and B*). Like CHIR-99021, silencing of both *GSK3A* and *GSK3B* augmented the expression of MAFB, with the simultaneous silencing of both *GSK3A* and *GSK3B* genes having a stronger effect (*Figure 4B*) and modulated the expression of 329 genes (*Figure 4C and D*). Of note, silencing of *GSK3A* and *GSK3B* reproduced the effects of CHIR-99021, as it strongly enhanced the expression of genes upregulated by the inhibitor in either GM-MØ or ex vivo isolated alveolar macrophages (*Figure 4E* and not shown). *GSK3A/B* knockdown triggered a significant enrichment in the expression of MAFB-dependent genes (*Figure 4F*), genes acquired along the monocyte-to-M-MØ differentiation (*Figure 4G*), and genes that mark M-MØ, while

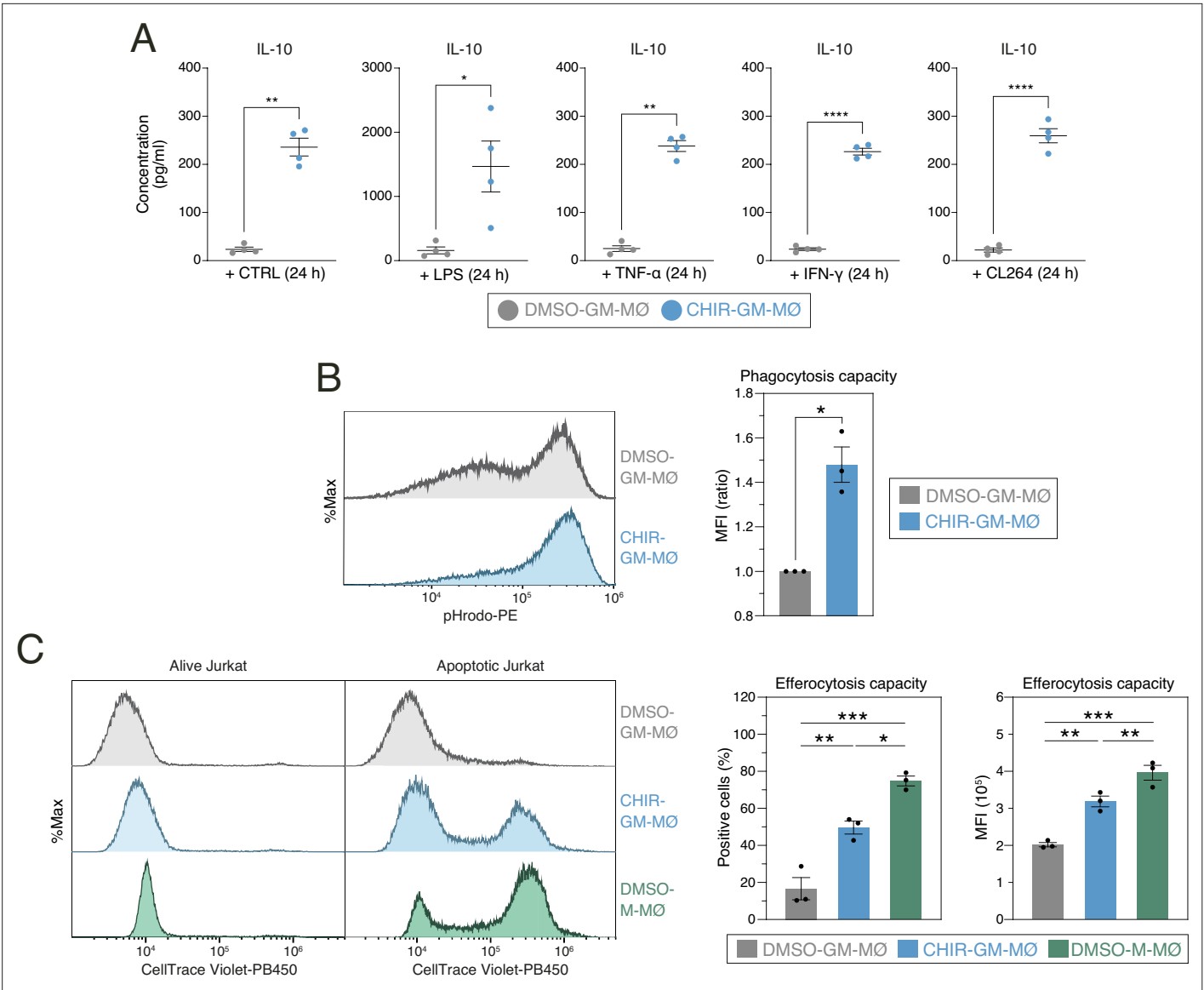

**Figure 3.** Functional consequences of GSK3 inhibition on monocyte-derived GM-MØ. (**A**) Production of IL-10 by untreated (-) or stimulated (LPS, TNF, IFNγ, or CL264) DMSO-GM-MØ and CHIR-GM-MØ, as determined by ELISA. Mean ± SEM of four independent samples are shown (paired Student's t-test: *, p<0.05; **, p<0.01; ***, p<0.005; ****, p<0.001). (**B**) Phagocytosis of pHRodo-labeled bacterial particles by DMSO-GM-MØ and CHIR-GM-MØ, as determined by flow cytometry. Mean ± SEM of three independent samples is shown (paired Student's t-test: *, p<0.05). A representative flow cytometry analysis is shown in the left panel. (**C**) Efferocytosis capacity of DMSO-GM-MØ, CHIR-GM-MØ, and DMSO-M-MØ, as determined by flow cytometry using staurosporine-induced CellTrace Violet-labeled apoptotic Jurkat cells. The percentage of positive cells and mean fluorescence intensity are shown. Mean ± SEM of three independent samples are shown (one-way ANOVA with Fisher's LSD test: *, p<0.05; **, p<0.01; ***, p<0.005). Representative flow cytometry histograms for CellTrace Violet emission of DMSO-GM-MØ, CHIR-GM-MØ, and DMSO-M-MØ are shown.

The online version of this article includes the following source data and figure supplement(s) for figure 3:

**Source data 1.** Concentration of IL10 by untreated (-) or stimulated (LPS, TNF, IFNγ, or CL264) DMSO-GM-MØ and CHIR-GM-MØ from four independent experiments and statistical analysis.

**Source data 2.** Phagocytosis of pHRodo-labeled bacterial particles.

**Source data 3.** Efferocytosis of apoptotic Jurkat cells.

**Figure supplement 1.** Oxygen consumption rate (OCR) profile of DMSO-GM-MØ and CHIR-GM-MØ monitored using the Seahorse Biosciences extracellular flux analyzer at the indicated time points.

**Figure supplement 2.** Extracellular acidification rate (ECAR), a proxy for the rate of lactate production, measured in DMSO-GM-MØ and CHIR-GM-MØ under basal conditions after the stimulation with 1 mM oligomycin and at the indicated time points.

*Figure 3 continued on next page*

*Figure 3 continued*

**Figure supplement 3.** Metabolic parameters obtained from the oxygen consumption rate (OCR) and extracellular acidification rate (ECAR) profiling after subtraction of the rotenone/antimycin-insensitive respiration.

reducing the expression of GM-MØ-specific genes (*Figure 4G*). Furthermore, in agreement with the effect of CHIR-99021, *GSK3A/B* knockdown enhanced the expression of the MoMac-VERSE macrophage clusters HES1_Mac (#2) and TREM2_Mac (#3) 2 (*Figure 4H*) as well as the gene sets that define pathogenic pro-fibrotic macrophage subsets in severe COVID-19 ([SPP1+, GSE145926] [*Liao et al., 2020*], MoAM3 [GSE155249] [*Grant et al., 2021*] or CD163+/LGMN+ MØ [EGAS00001005634] [*Wendisch et al., 2021*]) (*Figure 4I*, *Figure 4—figure supplement 1*). Conversely, silencing of *GSK3A* and *GSK3B* drastically reduced the expression of the gene sets that characterize tissue-resident lung alveolar macrophages in COVID-19, with the concomitant silencing of *GSK3A* and *GSK3B* again showing a stronger effect (*Figure 4I*). Altogether, these results support the relevance of GSK3α and GSK3β as relevant targets for human macrophage reprogramming.

## GSK3 inhibition redirects monocytes toward acquiring a gene profile reminiscent of anti-inflammatory M-MØ

Given the swift and profound anti-inflammatory impact observed in fully differentiated GM-MØ upon GSK3 inhibition or knockdown, we next delved into assessing whether modulation of GSK3 activity influences the differentiation capability of peripheral blood monocytes. To that end, monocytes were exposed to CHIR-99021, and the resulting cells (CHIR-Mon, *Figure 5A*) exhibited increased MAFB levels as early as 16 hr post-treatment (*Figure 5B*). Remarkably, the 16 hr exposure to the GSK3 inhibitor induced significant alterations in the expression of over 2000 genes ([log$_2$FC]>1; adjp<0.05) (*Figure 5C*) and sufficed to augment the expression of M-MØ-specific genes and genes upregulated during M-CSF-dependent monocyte-to-M-MØ differentiation (*Figure 5D*), including *CCL2*, *CD163*, and *IL10* (*Figure 5E*). Notably, the transcriptome of CHIR-Mon exhibited a significant enrichment in MAFB-dependent genes (*Figure 5*, *Figure 5—figure supplement 1*) and genes characterizing the MAFB+ pathogenic macrophages in severe COVID-19 (*Figure 5—figure supplement 2*), whereas it showed an under-representation of genes inhibited by MAFB (*Simón-Fuentes et al., 2023*; *Figure 5F*). Compared to DMSO, CHIR-99021 treatment upregulated the expression of 25% of M-MØ-specific genes, including MAFB-dependent genes like *CCL2*, *IL10,* and *CD163*, while downregulating 30% of GM-MØ-specific genes (*Figure 5G*), and this transcriptional effect was also evident in comparison with M-CSF-treated monocytes (*Figure 5—figure supplement 3*). To broaden the scope of these findings, we also examined the consequences of inhibiting GSK3 in monocytes exposed to a pathologic pro-inflammatory environment, namely synovial fluid from patients with active rheumatoid arthritis (RASF) (*Figure 5H*). Evaluation of CHIR-99021-treated monocytes in the presence of RASF revealed an enhanced expression of MAFB (*Figure 5I*) and MAFB-dependent genes such as *IL10* and *LGMN* (*Figure 5J*). Therefore, GSK3 inhibition in monocytes also leads to heightened MAFB expression, ultimately steering differentiation toward anti-inflammatory monocyte-derived macrophages.

## Inhibition of GSK3 in ex vivo isolated human alveolar macrophages promotes the acquisition of an M-MØ-like anti-inflammatory profile

To further validate the macrophage reprogramming potential of GSK3 inhibition, we next conducted experiments on ex vivo isolated human alveolar macrophages (AMØ, *Figure 6A*), whose development and gene expression profile is GM-CSF-dependent (*Guilliams et al., 2013*; *Trapnell et al., 2009*; *Huang et al., 2023*). To that end, we determined the transcriptome of three independent isolates of AMØ, purified from bronchoalveolar lavages (BALs), after exposure to CHIR-99021 for 24 hr (*Figure 6A*). A total of 865 differentially expressed genes ([log$_2$FC]>1; adjp<0.05) were identified, with 316 genes upregulated in CHIR-AMØ (*Figure 6B*). Further comparisons and gene ontology analysis revealed that GSK3 inhibition by CHIR-99021 (CHIR-AMØ) upregulated the gene sets that define pathogenic pro-fibrotic macrophage subsets in severe COVID-19 ([SPP1+, GSE145926] [*Liao et al., 2020*], MoAM3 [GSE155249] [*Grant et al., 2021*], or CD163+/LGMN+ MØ [EGAS00001005634] [*Wendisch et al., 2021*]) (*Figure 6C and D*; *Figure 6—figure supplement 1*). Conversely, CHIR-AMØ exhibited a reduced expression of the gene sets that characterize lung-resident AMØ in severe COVID-19

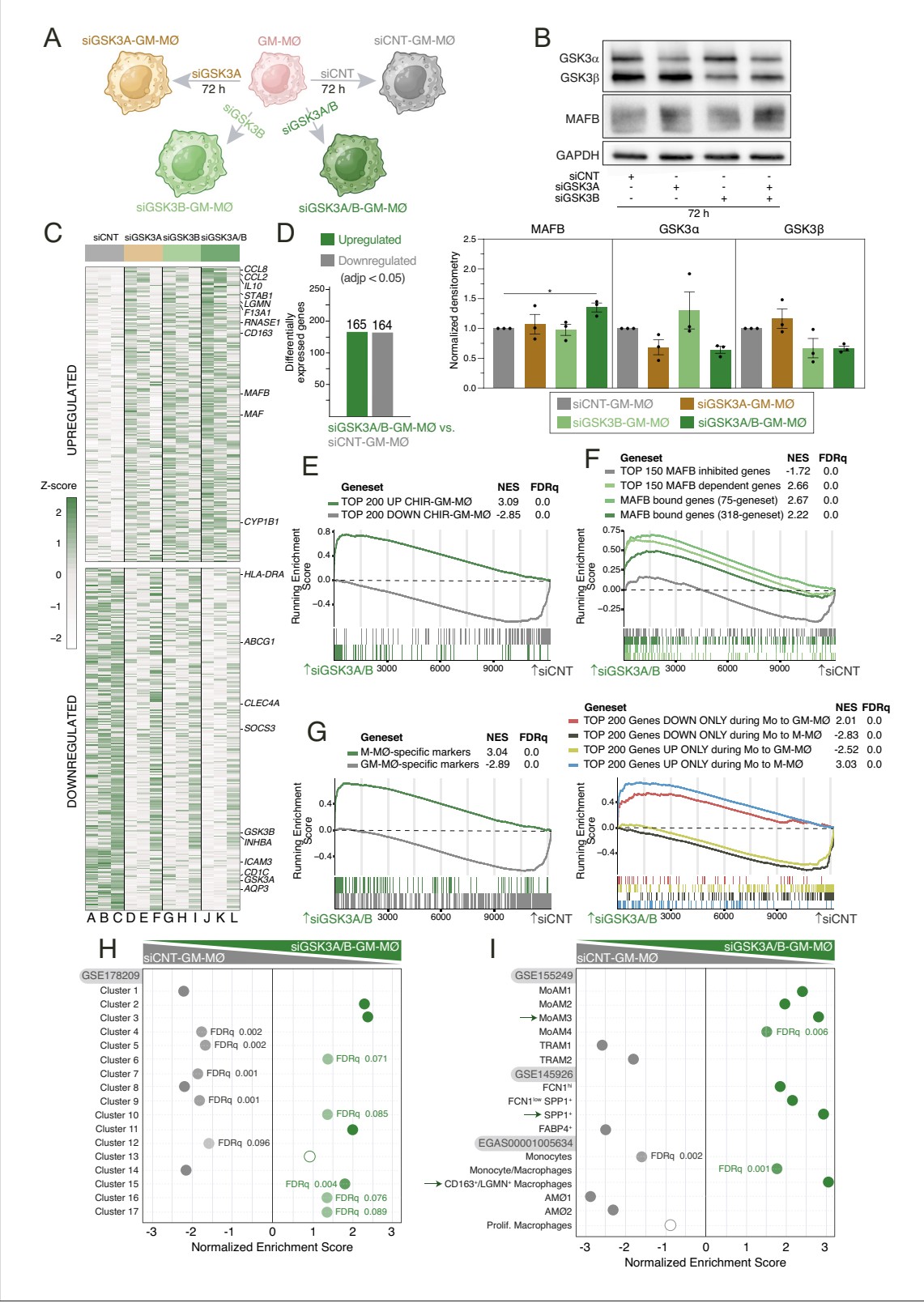

**Figure 4.** Transcriptional effects of GSK3 knockdown in monocyte-derived GM-MØ. (**A**) Schematic representation of the siRNA-mediated GSK3 knockdown procedure in monocyte-derived GM-MØ. Figure 4 A was created with BioRender.com. (**B**) Total GSK3α, GSK3β, and MAFB protein levels in monocyte-derived GM-MØ after siRNA-mediated *GSK3A* and/or *GSK3B* silencing, as determined by western blot (upper panel). GAPDH protein levels were determined as protein loading control. Mean ± SEM of GSK3β/GAPDH and GSK3α/GAPDH protein ratios from the three independent

*Figure 4 continued on next page*

*Figure 4 continued*

experiments are shown (lower panel) (one-way ANOVA with Dunnett's test: *, p<0.05). (**C**) Heatmap of the relative expression of the genes significantly altered after *GSK3A* and *GSK3B* knockdown in siCNT-GM-MØ (lanes A–C), siGSK3A-GM-MØ (lanes D–F), siGSK3B-GM-MØ (lanes G–I), siGSK3A/B-GM-MØ (lanes J–L). Representative genes are indicated. (**D**) Number of differentially expressed genes (adjp<0.05) between siGSK3A-GM-MØ, siGSK3B-GM-MØ, or siGSK3A/B-GM-MØ and siCNT-GM-MØ. Differential gene expression was assessed using DESeq2. (**E**) Gene set enrichment analysis (GSEA) of the genes whose expression is significantly modulated by CHIR-99021 in either GM-MØ (from *Figure 1*, this manuscript) or in ex vivo isolated alveolar macrophages (AMØ, see below in *Figure 6*) on the ranked comparison of the siGSK3A/B-GM-MØ vs. siCNT-GM-MØ transcriptomes. (**F**) GSEA of the indicated gene sets (from GSE68061) on the ranked comparison of the siGSK3A/B-GM-MØ vs. siCNT-GM-MØ transcriptomes. (**G**) Summary of GSEA of the indicated gene sets (from GSE188278) on the ranked comparison of siGSK3A/B-GM-MØ vs. siCNT-GM-MØ. (**H**) Summary of GSEA of the gene sets that define the tissue-resident monocyte and macrophage states (MoMac-VERSE) *Mulder et al., 2021* on the ranked comparison of the siGSK3A/B-GM-MØ vs. siCNT-GM-MØ transcriptomes. FDRq values are indicated only if FDRq>0.0; empty dots, not significant. (**I**) Summary of GSEA of the gene sets that characterize the macrophage subsets identified in severe COVID-19 (*Wendisch et al., 2021*; *Liao et al., 2020*; *Grant et al., 2021*) on the ranked comparison of siGSK3A/B-GM-MØ vs. siCNT-GM-MØ. FDRq values are indicated only if FDRq>0.0; empty dots, not significant.

The online version of this article includes the following source data and figure supplement(s) for figure 4:

**Source data 1.** PDF file containing original western blots for *Figure 4B*, indicating the relevant bands.

**Source data 2.** Original files for western blot analysis displayed in *Figure 4B*.

**Source data 3.** Protein ratios of GSK3β/GAPDH and GSK3α/GAPDH in monocyte-derived GM-MØ after siRNA-mediated GSK3A and/or GSK3B silencing from the three independent experiments and statistical analysis.

**Figure supplement 1.** Summary of gene set enrichment analysis (GSEA) of the gene sets that characterize the macrophage subsets identified in severe COVID-19 (*Wendisch et al., 2021*; *Liao et al., 2020*; *Grant et al., 2021*) on the ranked comparison of siGSK3A-GM-MØ vs. siCNT-GM-MØ (square symbols), siGSK3B-GM-MØ vs. siCNT-GM-MØ (star symbols), and siGSK3A/B-GM-MØ vs. siCNT-GM-MØ.

([AMØ1+AMØ2 clusters from EGAS00001005634] [*Wendisch et al., 2021*], FABP4+ from GSE145926 [*Liao et al., 2020*], TRAM1 from GSE155249 [*Grant et al., 2021*]) (*Figure 6D and E*; *Figure 6—figure supplement 2*), including key AMØ-specific genes like *FABP4* and *MARCO* (*Figure 6E*), and similar results were seen upon analysis of AMØ-specific gene sets from other studies (*Grant et al., 2021*; *Melms et al., 2021*). Like in the case of CHIR-GM-MØ, CHIR-AMØ exhibited an over-representation of M-MØ-specific genes and genes acquired along the monocyte-to-M-MØ differentiation (GSE188278) (*Figure 6—figure supplement 3*). Consistent with these findings, CHIR-AMØ exhibited higher expression of MAFB (*Figure 6F*), whose increase correlated with an augmented secretion of Legumain, CCL2, and IL-10 (*Figure 6G*), although the latter was only seen in two samples, probably reflecting heterogeneity in primary cell responses. In contrast, lower levels of Activin A were detected in CHIR-AMØ-conditioned medium (*Figure 6G*). Remarkably, and in agreement with the shift in macrophage subsets that takes place in severe COVID-19 lungs (loss of AMØ, appearance of monocyte-derived pro-fibrotic macrophages) (*Vega et al., 2020*; *Wendisch et al., 2021*; *Liao et al., 2020*; *Grant et al., 2021*), the level of phosphorylated Ser[21]-GSK3α and Ser[9]-GSK3β was very significantly augmented in lung macrophages (CD68+) from severe COVID-19 patients, an increase that correlated with a robust enhancement of the macrophage expression of both MAFB and the MAFB-dependent protein CD163 (*Figure 6H*). These results collectively demonstrate that macrophage reprogramming through GSK3 inhibition can also be achieved on in vivo isolated macrophages, leading to enhanced expression of MAFB and MAFB-dependent genes, the acquisition of an anti-inflammatory transcriptional profile, and the loss of the signature of lung-resident human AMØ.

## GSK3- and MAFB-dependent genes are differentially expressed between human lung alveolar and interstitial macrophages

To assess whether GSK3 inhibition also modulates the phenotypic differences between human lung macrophage subsets in a physiological setting, we next took advantage of the available information on human healthy lungs obtained by single-cell RNA sequencing (*Morse et al., 2019*). Re-analysis of the single-cell RNA sequencing data (*Morse et al., 2019*: *Figure 7A*; *Figure 7—figure supplements 1–4*) revealed several clusters expressing macrophage-associated markers (*CD163, FABP4, LYVE1, FCN1*), and whose subsequent re-clustering identified cell subsets expressing previously defined markers for infiltrating monocytes (Cluster 4/7/12: *FCN1*) (*Wendisch et al., 2021*), alveolar macrophages (AMØ) (Clusters 0/2/5/13: *FABP4 TYMS*-negative; Cluster 9: *MARCO, INHBA*; Cluster 11: *PPARG*), proliferating AMØ (Cluster 15: *TYMS, MKI67, TOP2A, NUSAP1*) (*Wendisch et al., 2021*), and interstitial macrophages (IMØ) that variably express markers for tissue-resident macrophages

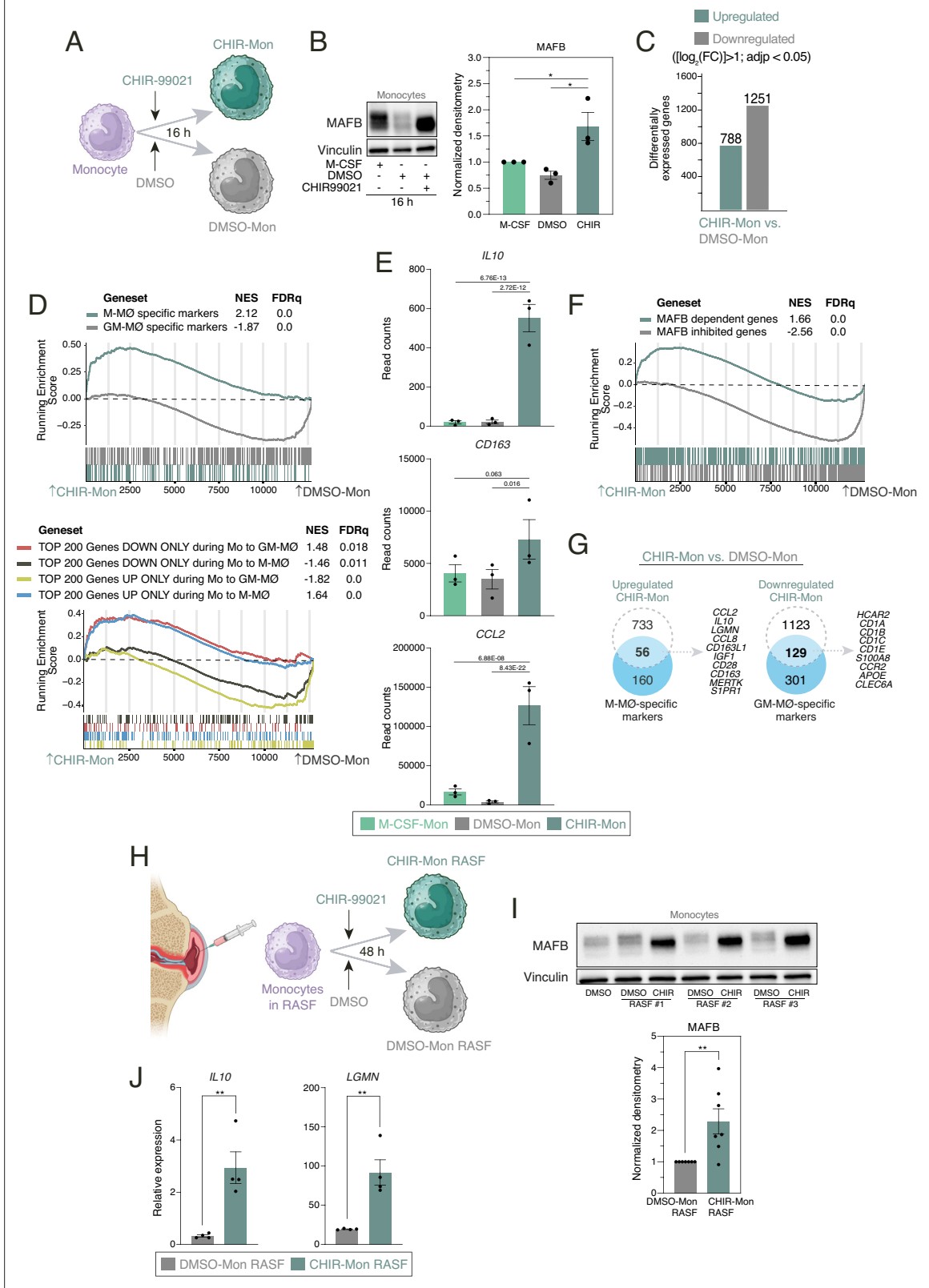

**Figure 5.** Transcriptional effects of GSK3 inhibition on human peripheral blood monocytes. (**A**) Schematic representation of the exposure of human monocytes to CHIR-99021 or DMSO for 16 hr. Figure 5A was created with BioRender.com. (**B**) MAFB protein levels in monocytes exposed for 16 hr to DMSO, CHIR-99021, or M-CSF, as determined by western blot (left panel). Vinculin protein levels were determined as protein loading control. Mean ± SEM of the MAFB/Vinculin protein ratio from the three independent experiments is shown (right panel) (one-way ANOVA with Fisher's LSD test: ***,

*Figure 5 continued on next page*

*Figure 5 continued*

p<0.005). (**C**) Number of differentially expressed genes ([log2FC]>1; adjp<0.05) between DMSO-Mon and CHIR-Mon. Differential gene expression was assessed using DESeq2. (**D**) Gene set enrichment analysis (GSEA) of the indicated gene sets (from GSE68061, upper panel; from GSE188278, lower panel) on the ranked comparison of the DMSO-Mon and CHIR-Mon transcriptomes. NES and FDRq values are indicated in each case. (**E**) Relative mRNA levels of the indicated genes in monocytes exposed for 16 hr to DMSO, CHIR-99021, or M-CSF, as determined by RNA-Seq on three independent samples (GSE256538). Adjp of the comparison, shown in each case, was assessed using DESeq2. (**F**) GSEA of MAFB-regulated gene sets (from GSE155719) on the ranked comparison of the DMSO-Mon and CHIR-Mon transcriptomes. NES and FDRq values are indicated in each case. (**G**) Overlap between the genes upregulated ([log2FC]>1; adjp<0.05) in CHIR-Mon (relative to DMSO-Mon) and M-MØ-specific marker genes (from GSE68061) (left panel) and genes downregulated ([log2FC]>1; adjp<0.05) in CHIR-Mon (relative to DMSO-Mon) and GM-MØ-specific marker genes (from GSE68061) (right panel), with indication of some of the overlapping genes. (**H**) Schematic representation of the exposure of human monocytes to synovial fluid from rheumatoid arthritis patients (RASF) for 48 hr in the presence or absence of CHIR-99021. Figure 5H created with BioRender.com. (**I**) MAFB protein levels in monocytes exposed to three independent samples of RASF in the presence of DMSO or CHIR-99021, as determined by western blot (upper panel). Vinculin protein levels were determined as protein loading control. Mean ± SEM of the MAFB/Vinculin protein ratio from seven independent experiments using three monocyte preparations and three unrelated RASF (lower panel) (paired Student's t-test: **, p<0.01). (**J**) Relative expression of *IL10* and *LGMN* in monocytes exposed to RASF in the presence of DMSO or CHIR-99021, as determined by RT-PCR. Mean ± SEM of the results from the three independent samples are shown (paired Student's t-test: **, p<0.01).

The online version of this article includes the following source data and figure supplement(s) for figure 5:

**Source data 1.** PDF file containing original western blots for *Figure 5B*, indicating the relevant bands.

**Source data 2.** Original files for western blot analysis displayed in *Figure 5B*.

**Source data 3.** Protein ratios of MAFB/Vinculin in monocytes exposed for 16 hr to DMSO, CHIR-99021, or M-CSF from the three independent experiments and statistical analysis.

**Source data 4.** Relative mRNA levels of the indicated genes in monocytes exposed for 16 hr to DMSO, CHIR-99021, or M-CSF on three independent experiments.

**Source data 5.** DF file containing original western blots for *Figure 5I*, indicating the relevant bands.

**Source data 6.** Original files for western blot analysis displayed in *Figure 5I*.

**Source data 7.** Protein ratios of MAFB/Vinculin in monocytes exposed to synovial fluid from rheumatoid arthritis patients (RASF) in the presence of DMSO or CHIR-99021 from seven independent experiments and statistical analysis.

**Source data 8.** Relative expression of *IL10* and *LGMN* in monocytes exposed to synovial fluid from rheumatoid arthritis patients (RASF) in the presence of DMSO or CHIR-99021 from four independent experiments and statistical analysis.

**Figure supplement 1.** Gene set enrichment analysis (GSEA) of bona fide MAFB-bound genes ('75-genelist') (*Simón-Fuentes et al., 2023*) on the ranked comparison of the transcriptomes of CHIR-Mon vs. DMSO-Mon.

**Figure supplement 2.** Summary of gene set enrichment analysis (GSEA) of the gene sets that characterize the macrophage subsets identified in severe COVID-19 (*Wendisch et al., 2021*; *Liao et al., 2020*; *Grant et al., 2021*) on the ranked comparison of the transcriptomes of CHIR-Mon vs. DMSO-Mon.

**Figure supplement 3.** Overlap between the genes upregulated ([log2FC]>1; adjp<0.05) in CHIR-Mon (relative to M-CSF-Mon) and M-MØ-specific marker genes (left panel) and genes downregulated ([log2FC]>1; adjp<0.05) in CHIR-Mon (relative to M-CSF-Mon) and GM-MØ-specific marker genes (right panel), with indication of some shared genes.

---

like *LYVE1*, *RNASE1*, and *LGMN* (Clusters 1/3/6/8/10/14) (*Dick et al., 2022*; *Wendisch et al., 2021*; *Morse et al., 2019*: *Figure 7B*; *Figure 7—figure supplements 5–7*). The comparison of IMØ and AMØ subsets revealed that the genes upregulated upon inhibition of GSK3 include numerous MAFB-dependent genes (e.g. *MAFB*, *IL10*, *LGMN*, *SPP1*, *CCL2*) and showed a higher expression in IMØ (*Figure 7C*). On the other hand, the genes downregulated by CHIR-99021 were preferentially or exclusively expressed in AMØ, including *INHBA*, *GSN*, *FBP1*, *PPARG*, *AXL*, and *AQP3* (*Figure 7C*). Therefore, GSK3 inhibition prompts macrophages toward the acquisition of a gene profile that resembles that of IMØ, while provoking the loss of the AMØ gene signature. Such a conclusion was also reached upon analysis of pseudo bulk RNA-Seq data from GSE128033 (*Morse et al., 2019*). Indeed, numerous MAFB-dependent and GSK3-regulated genes were found within the 712 genes that are differentially expressed ([log$_2$FC]>1; adjp<0.05) between IMØ and AMØ (*Figure 7D*). Further GSEA evidenced that GSK3 inhibition in either GM-MØ (CHIR-GM-MØ) (*Figure 7E*) or AMØ (CHIR-AMØ) (*Figure 7F*) very significantly enhanced the expression of MAFB-dependent IMØ-specific genes (as defined by *Morse et al., 2019*, or by *Li, 2022*), while simultaneously impairing the expression of genes characterizing human AMØ (*Figure 7E and F*). More importantly, a similar effect was observed after *GSK3A/B* silencing in GM-MØ, which yielded an enhanced expression of IMØ-specific genes and a downregulation of the gene set that defines AMØ (*Figure 7G*). Altogether, these results indicate that inhibiting GSK3 shifts the gene profile of alveolar macrophages toward that of interstitial

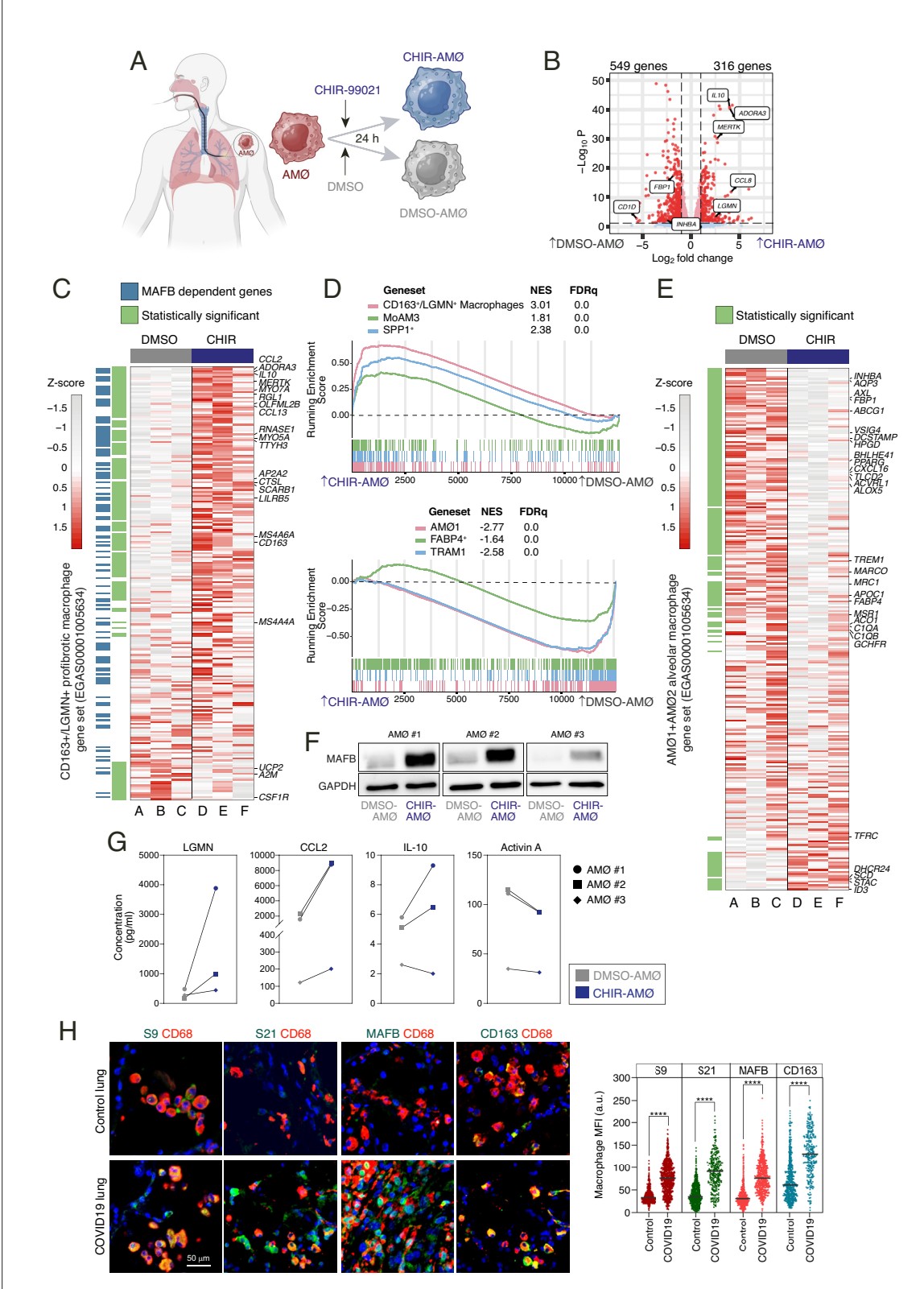

**Figure 6.** Transcriptional and phenotypic effects of GSK3 inhibition on ex vivo isolated human alveolar macrophages. (**A**) Schematic representation of the exposure of ex vivo isolated human alveolar macrophages to CHIR-99021 or DMSO for 24 hr. Figure 6A created with BioRender.com. (**B**) Volcano plot depicting the differentially expressed genes between CHIR-AMØ and DMSO-AMØ. (**C**) Heatmap of the relative expression of the genes within the CD163+/LGMN+ pro-fibrotic monocyte-derived macrophage cluster (from EGAS00001005634) (*Wendisch et al., 2021*) in DMSO-AMØ (lanes **A–C**) and

*Figure 6 continued on next page*

*Figure 6 continued*

CHIR-AMØ (lanes **D–F**). Differentially expressed between CHIR-AMØ and DMSO-AMØ are shown in green, and MAFB-dependent genes (GSE155719) are shown in blue. Representative genes are indicated. (**D**) (Upper panel) Gene set enrichment analysis (GSEA) of the gene sets that characterize the monocyte-derived pro-fibrotic macrophage subsets identified in severe COVID-19, namely CD163+/LGMN+ (EGAS00001005634) (*Wendisch et al., 2021*), MoAM3 (GSE155249) (*Grant et al., 2021*), and SPP1+ (GSE145926) (*Liao et al., 2020*), on the ranked comparison of the transcriptomes of CHIR-AMØ vs. DMSO-AMØ. NES and FDRq values are indicated in each case. (Lower panel) GSEA of the gene sets that characterize tissue-resident alveolar macrophages in severe COVID-19, namely AMØ1 (EGAS00001005634) (*Wendisch et al., 2021*), TRAM1+ (GSE155249) (*Grant et al., 2021*), and FABP4+ (GSE145926) (*Liao et al., 2020*) on the ranked comparison of the transcriptomes of CHIR-AMØ vs. DMSO-AMØ. (**E**) Heatmap of the relative expression of the genes within the alveolar macrophage signature (including all the genes overexpressed in the AMØ1 and AMØ2 macrophages clusters from EGAS00001005634) (*Wendisch et al., 2021*) in DMSO-AMØ (lanes A–C) and CHIR-AMØ (lanes D–F). Differentially expressed between CHIR-AMØ and DMSO-AMØ are indicated in green, and selected genes are shown. (**F**) MAFB protein levels in three independent samples of ex vivo isolated AMØ exposed to DMSO or CHIR-99021 for 24 hr, as determined by western blot. GAPDH protein levels were determined as protein loading control. (**G**) Production of the indicated soluble factors by three independent preparations of DMSO-AMØ and CHIR-AMØ, as determined by ELISA. (**H**) Expression of inactive GSK3 (S9, pSer$^9$-GSK3β; S21, pSer$^{21}$-GSK3α), MAFB, and CD163 in human lung macrophages. Representative human lung tissues from COVID (n=2) and control (n=3) patients co-stained for CD68 (macrophage marker, red), Ser$^{21}$-phosphorylated GSK3α, Ser$^9$-phosphorylated GSK3β, MAFB or CD163 (green), and DAPI (nuclei, blue), as indicated (scale bar, 50 μm) (left panel). Plots show mean fluorescence intensity (MFI) (in arbitrary units, a.u.) of single-cell CD68+ macrophages stained for each antibody (right panel) (paired Student's t-test: ****, $p < 0.001$).

The online version of this article includes the following source data and figure supplement(s) for figure 6:

**Source data 1.** PDF file containing original western blots for *Figure 6F*, indicating the relevant bands.

**Source data 2.** Original files for western blot analysis displayed in *Figure 6F*.

**Source data 3.** Concentration of CCL2, IL10, LGMN, and Activin A in DMSO-AMØ and CHIR-AMØ from three independent preparations and statistical analysis.

**Source data 4.** Mean fluorescence intensity of single-cell CD68+ macrophages stained with antibodies for inactive GSK3, MAFB, and CD163 in human lung macrophages from two independent COVID-19 samples.

**Figure supplement 1.** Overlap between the genes upregulated or downregulated in CHIR-AMØ (|log2FC|>1 and adjp<0.05) and the CD163+/LGMN+ pro-fibrotic macrophage cluster defined in EGAS00001005634 (*Wendisch et al., 2021*) (left panel) or the gene sets of MAFB-dependent (middle panel) and MAFB-inhibited (right panel) genes defined in GSE155719 (*Simón-Fuentes et al., 2023*).

**Figure supplement 2.** Overlap between the genes upregulated or downregulated in CHIR-AMØ (|log2FC|>1 and adjp<0.05, or just adjp<0.05) and the alveolar macrophage signature as defined in EGAS00001005634 (AMØ1 and AMØ2 clusters) (*Wendisch et al., 2021*) (left panel), GSE155249 (TRAM1 and TRAM2 clusters) (*Grant et al., 2021*) (middle panel), or in *Melms et al., 2021* (right panel).

**Figure supplement 3.** Summary of gene set enrichment analysis (GSEA) of the indicated gene sets (from GSE68061 and GSE188278) on the ranked comparison of the DMSO-AMØ and CHIR-AMØ transcriptomes.

macrophages (or monocyte-derived macrophages), and imply that the activation status of GSK3 significantly influences the specification of lung-resident macrophages (AMØ vs. IMØ).

## Discussion

Our previous investigations have demonstrated that the macrophage reprogramming effect of LXR modulators (*de la Aleja et al., 2023*; *González de la Aleja et al., 2022*) and methotrexate (*Ríos et al., 2023*) coincide with alterations in the protein levels of MAFB. Similarly, intravenous immunoglobulins have been shown to alter the inflammatory state of macrophages and modulate MAFB levels (*Domínguez-Soto et al., 2018*). Building upon these findings and others, we have explored whether modulating the activity of GSK3, a primary regulator of MAFB protein levels and activity, is sufficient to induce a shift in the inflammatory profile of human macrophages. Our results indicate that GSK3 inhibition impairs the expression of genes associated with GM-CSF-dependent differentiation of monocyte-derived macrophages while concurrently promoting the acquisition of genes and functions associated with M-CSF-dependent monocyte-derived macrophages, particularly those regulated by MAFB. The physiological relevance of these findings is underscored by the contrasting effect that GSK3 inhibition has on genes that define lung-resident alveolar macrophages (decrease) vs. monocyte-derived macrophages recruited into the lung (increase) during inflammatory responses. Thus, modulation of GSK3 activity emerges as a plausible strategy for macrophage reprogramming in therapeutic interventions.

Therapeutic approaches aimed at re-educating tumor-associated macrophages into immunostimulatory cells (*Molgora and Colonna, 2021*) may involve targeting the macrophage PI3Kγ intracellular

signaling pathway (*Zheng and Pollard, 2016*), which inhibits NFκB activation through Akt and mTOR (*Kaneda et al., 2016*), thereby inducing an anti-inflammatory program that promotes immune suppression. Indeed, macrophage PI3Kγ inactivation stimulates and prolongs NFκB activation, promoting an immunostimulatory transcriptional program (*Kaneda et al., 2016*). The ability of PI3Kγ to toggle between immune stimulation and suppression aligns with our findings on GSK3 inhibition, as PI3Kγ inactivates GSK3 via phosphorylation of $Ser^9$ (GSK3β) and $Ser^{21}$ (GSK3α) (*Eychène et al., 2008*). Therefore, GSK3 inhibition may contribute to the anti-inflammatory and immunosuppressive effects of PI3Kγ inactivation by enhancing MAFB expression. Moreover, MAFB might even participate in the macrophage reprogramming consequences of mTOR modulation (*Byles et al., 2013*; *Do et al., 2023*; *Zhang et al., 2023*) due to the known reciprocal interactions between GSK3 and mTOR (*Evangelisti et al., 2020*). Hence, since the phosphorylation of MAFB by GSK3 requires a prior 'priming' phosphorylation event (*Eychène et al., 2008*), identification of the MAFB priming kinase(s) emerges as a potentially relevant effort in the search for additional targets for macrophage reprogramming.

The significant impact of GSK3 inhibition by CHIR-99021 on the macrophage transcriptional profile mirrors the effects observed with other GSK3 inhibitors on the differentiation of monocyte-derived dendritic cells (MDDCs) (*Rodionova et al., 2007*). Specifically, GSK3 inhibitors like LiCl, SB415286, or SB216763 redirect the GM-CSF+IL-4-dependent development of MDDC toward the generation of macrophage-like cells (*Rodionova et al., 2007*). Considering that MAFB expression is higher in monocyte-derived macrophages than in dendritic cells (*Bakri et al., 2005*), it is reasonable to infer that GSK3 inhibitors also boost MAFB expression in cells differentiated in the presence of GM-CSF+IL-4, thereby supporting the modulation of GSK3 activity as a tool to alter lineage determination in myeloid cells.

The reprogramming effect of GSK3 inhibition on ex vivo isolated alveolar macrophages bears significant therapeutic implications, particularly in severe COVID-19 cases where pro-fibrotic monocyte-derived macrophages replace GM-CSF-dependent tissue-resident alveolar macrophages (*Wendisch et al., 2021*; *Liao et al., 2020*; *Delorey et al., 2021*). Our observations reveal that exposure of ex vivo isolated alveolar macrophages to CHIR-99021 leads to the loss of the alveolar macrophage GM-CSF-dependent gene profile and the acquisition of a MAFB-dependent pro-fibrotic transcriptional landscape resembling that of M-CSF-dependent monocyte-derived macrophages. Given that MAFB serves as a negative regulator of GM-CSF signaling (*Koshida et al., 2015*), downregulation of MAFB expression through modulation of GSK3 activity emerges as a rationale strategy to re-direct pathogenic monocyte-derived macrophages toward the acquisition of a lung-resident alveolar macrophage-like profile during severe COVID-19. This approach aligns with the proposed therapeutic use of GM-CSF in COVID-19 (*Lang et al., 2020*; *Bosteels et al., 2022*), as GM-CSF would impede or limit the MAFB-dependent upregulation of pro-fibrotic and neutrophil-attracting factors observed in severe COVID-19 (*Simón-Fuentes et al., 2023*).

## Materials and methods
### Generation of human monocyte-derived macrophages in vitro and treatments

Human peripheral blood mononuclear cells (PBMCs) were isolated from buffy coats from anonymous healthy donors over a Lymphoprep (Nycomed Pharma) gradient according to standard procedures. Monocytes were purified from PBMC by magnetic cell sorting using anti-CD14 microbeads (Miltenyi Biotec). Monocytes (>95% CD14+ cells) were cultured at $0.5 \times 10^6$ cells/ml in Roswell Park Memorial Institute (RPMI 1640, Gibco) medium supplemented with 10% fetal bovine serum (FBS, Biowest) (complete medium) for 7 days in the presence of 1000 U/ml GM-CSF or 10 ng/ml M-CSF (ImmunoTools) to generate GM-CSF-polarized macrophages (GM-MØ) or M-CSF-polarized macrophages (M-MØ), respectively (*Cuevas et al., 2017*). Cytokines were added every 2 days, and cells were maintained at 37°C in a humidified atmosphere with 5% $CO_2$ and 21% $O_2$. Alveolar macrophages (AMØ) were obtained from remains of BAL of patients undergoing bronchoscopy for diagnostic purposes under a protocol approved by the Internal Review Board of Instituto de Investigación Hospital 12 de Octubre (Reference TP24/0183, including informed consents). BAL procedure was performed with a flexible bronchoscope and a total volume of 150 ml of sterile isotonic saline solution at 37°C. BAL fluid fractions were maintained at 4°C, and cellular debris removed using a 40 µm cell strainer. BAL cells were

washed with PBS, centrifuged, and resuspended in complete medium containing 100 U/mL penicillin and 100 µg/ml streptomycin (#15140-122, Gibco), 50 µg/ml gentamicin (#G1397, Sigma-Aldrich), and 2.5 µg/ml amphotericin B (#A2942, Sigma-Aldrich). The cells were seeded at $6–8×10^5$ cells per well in 12-well plates for 1 hr and washed extensively to remove non-adherent cells. Finally, 2 ml of complete medium with antibiotics was added to each well, and the adherent cells incubated for 16–18 hr before treatments. More than 95% of adherent BAL cells were identified as AMØ according to morphology and phenotypic analysis. When indicated, monocytes, GM-MØ, or AMØ were exposed to the GSK3 inhibitors CHIR99021 (10 µM), SB-216763 (10 µM), or LiCl (10 mM), using DMSO as control. For macrophage activation, cells were treated with either 100 ng/ml CL264 (InvivoGen), 10 ng/ml *E. coli* 0111:B4 lipopolysaccharide (LPS-EB Ultrapure, Invivogen), TNF (20 ng/ml, ImmunoTools), or IFNγ (5 ng/ml, ImmunoTools). Human cytokine production was measured in M-MØ culture supernatants using commercial ELISA (CCL2 [BD Biosciences], IL-10, CCL18, LGMN, SPP1 [R&D Systems] and Activin A [BD Biosciences]) and following the procedures supplied by the manufacturers.

## siRNA transfection

GM-MØ ($1×10^6$ cells) were transfected with a human *GSK3A*-specific and/or *GSK3B*-specific siRNA (50 nM) (Dharmacon) using HiPerFect (QIAGEN). Silencer Select Negative Control No. 2 siRNA (siCtrl, 50 nM) (Dharmacon) was used as negative control siRNA. Six hours after transfection, cells were either allowed to recover from transfection in complete medium (66 hr) and lysed. Knockdown of GSK3α/β was confirmed by western blot.

## Quantitative real-time RT-PCR

Total RNA was extracted using the total RNA and protein isolation kit (Macherey-Nagel). RNA samples were reverse-transcribed with High-Capacity cDNA Reverse Transcription reagents kit (Applied Biosystems) according to the manufacturer's protocol. Real-time quantitative PCR was performed with LightCycler 480 Probes Master (Roche Life Sciences) and Taqman probes on a standard plate in a Light Cycler 480 instrument (Roche Diagnostics). Gene-specific oligonucleotides (*IL10*: Forward, 5′-TCACTCATGGCTTTGTAGATGC-3′, and Reverse, 5′- GTGGAGCAGGTGAAGAATGC-3′; *LGMN*: Forward, 5′-GAACACCAATGATCTGGAGGA-3′, and Reverse, 5′-GGAGACGATCTTACGCACTGA-3′) were designed using the Universal ProbeLibrary software (Roche Life Sciences). Results were normalized to the expression level of the endogenous reference genes (*TBP*, *HPRT1*) and quantified using the ΔΔCT (cycle threshold) method.

## Western blot

Cell lysates were subjected to SDS-PAGE (50 µg unless indicated otherwise) and transferred onto an Immobilon-P polyvinylidene difluoride membrane (Millipore). After blocking the unoccupied sites with 5% non-fat milk diluted in Tris-Buffered Saline plus Tween 20 (TBS-T), protein detection was carried out with antibodies against MAFB (HPA005653, Sigma-Aldrich), total GSK3α (#4337, Cell Signaling Technology), total GSK3β (#27C10, Cell Signaling Technology), total GSK3 (#5676, Cell Signaling Technology), p-Ser$^9$-GSK3β (#5558, Cell Signaling Technology), p-Ser$^{21}$-GSK3α (#9316, Cell Signaling Technology), p-Tyr$^{279}$/P-Tyr$^{216}$-GSK3α/β (#05-413, Sigma-Aldrich), CD163 (#MCA1853, Bio-Rad), MAF (sc-518062, Santa Cruz Biotechnology), IL7R (sc-514445, Santa Cruz Biotechnology) and FOLR2 (clone 94b, kindly provided by Dr. Takami Matsuyama) (*Nagayoshi et al., 2005*), and Vinculin (#V9131, Sigma-Aldrich) or GAPDH (sc-32233, Santa Cruz Biotechnology) as protein loading controls. Quimioluminiscence was detected in a Chemidoc Imaging system (Bio-Rad) using SuperSignal West Femto (Thermo Fisher Scientific).

## RNA-Seq and data analysis

RNA isolated from GM-MØ exposed to DMSO or the GSK3β inhibitor CHIR-99021 (10 µM) for 48 hr, monocytes exposed to CHIR99021 or DMSO, ex vivo isolated AMØ exposed to CHIR99021 or DMSO, or GM-MØ transfected with siRNA specific for *GSK3A* and/or *GSK3B*, or a control siRNA, were subjected to sequencing on a BGISEQ-500 platform (https://www.bgitechsolutions.com). RNA-Seq data were deposited in the Gene Expression Omnibus (http://www.ncbi.nlm.nih.gov/geo/) under accession GSE256538 (monocytes exposed to CHIR99021 or DMSO), GSE256208 (GM-MØ exposed to CHIR99021 or DMSO), GSE262463 (AMØ exposed to CHIR99021 or DMSO), or GSE266236

(GM-MØ transfected with siRNA specific for *GSK3A*, *GSK3B,* or a control siRNA). Low-quality reads and reads with adaptors or unknown bases were filtered to get the clean reads. Sequences were mapped to GRCh38 genome using HISAT2 (*Kim et al., 2019*) or Bowtie2 (*Langmead and Salzberg, 2012*), and clean reads for each gene were calculated using htseq-count (*Anders et al., 2015*) and the RSEM software package (*Li and Dewey, 2011*). Differential gene expression was assessed by using the R package DESeq2. Differentially expressed genes were analyzed for annotated gene sets enrichment using ENRICHR (http://amp.pharm.mssm.edu/Enrichr/) (*Kuleshov et al., 2016*; *Chen et al., 2013*), and enrichment terms considered significant with a Benjamini-Hochberg-adjusted p-value<0.05. For GSEA (http://software.broadinstitute.org/gsea/index.jsp) (*Subramanian et al., 2005*), gene sets available at the website, as well as gene sets generated from publicly available transcriptional studies (https://www.ncbi.nlm.nih.gov/gds), were used. The datasets used throughout the manuscript (either reported here for the first time or previously published) are listed and described in Table S1.

## Re-analysis of single-cell RNA-Seq data

Single-cell RNA-Seq data from eight healthy lung samples were obtained from GSE128033 (*Morse et al., 2019*) and further analyzed using R programming language. A total of 5,898,240 cells were filtered down to 21,310 according to the following criteria: (1) number of genes per cell (nFeature, >200 and <6000); (2) Unique Molecular Identifiers (nCount, >1000); and (3) % of mitochondrial genes (<15%). Data were then normalized using LogNormalize with a scale factor of 10,000 and the top 2000 features were identified with the vst method using Seurat v.5.0.1. The resulting Seurat object was converted into an Anndata object using sceasy v.0.0.7, and the integration was processed using scVI v.1.0.4 and Scanpy v.1.9.8 in a Python environment. The trained model was re-converted into a Seurat object. The Seurat FindNeighbors function was used with the scVI reduction to obtain the resulting clusters with a resolution of 1.2. To focus on macrophages and monocytes, macrophage (*CD163*, *FABP4*, *LYVE1* as markers) and monocyte (*FCN1* as marker) clusters were identified and re-integrated using scVI v.1.0.4 in a Python environment and later re-clustered in Seurat v.5.0.1 with a resolution of 0.6.

## Phagocytosis assay

Phagocytosis ability was assessed by flow cytometry using pHrodo Red *E. coli* BioParticles Conjugates (Thermo Fisher), following the procedures recommended by the manufacturer. Macrophages were cultured in a 24-well plate and exposed to pHrodo bioparticles for 60 min at 37°C/5% $CO_2$. Cells were then harvested and assessed by flow cytometry.

## Efferocytosis assay

Jurkat cells were cultured in RPMI 1640 medium without FCS for 16 hr and then treated with staurosporine (0.5 µg/ml, SIGMA), followed by an incubation at 37°C for 3 hr. Staurosporine treatment yielded a population of 83% annexin V + cells. Apoptotic cells (ACs) were resuspended at a concentration of $1 \times 10^6$ cells/ml and labeled with CellTrace Violet reagent (0.5 µM, Invitrogen, Thermo Scientific) for 20 min. For efferocytosis, macrophages were cultured with labeled AC (ratio 1:4) in p24 plates during 1 hr at 37°C. After 1 hr, macrophages were rinsed with PBS to remove unbound AC and detached with PBS 5 mM EDTA and pelleted by centrifugation before analyzing by flow cytometry.

## Bioenergetics profile

The XF24 extracellular flux analyzer (Seahorse Biosciences, North Billerica, MA, USA) was used to determine the bioenergetic profile of intact cells. Briefly, cells were seeded (200,000 cells/well) in XF24 plates (Seahorse Biosciences) and allowed to recover for 24 hr. Cells were then incubated in bicarbonate-free DMEM (Sigma-Aldrich) supplemented with 11.11 mM glucose, 2 mM L-glutamine, 1 mM pyruvate, and 2% FBS (Sigma-Aldrich) in a $CO_2$-free incubator for 1 hr. The oxygen consumption rate (OCR) and extracellular acidification rate (ECAR), a proxy for lactate production, were recorded to assess the mitochondrial respiratory activity and glycolytic activity, respectively. After four measurements under basal conditions, cells were treated sequentially with 1 µM oligomycin, 0.6 µM carbonyl cyanide *p*-(trifluoromethoxy)phenylhydrazone (FCCP), 0.4 µM FCCP, and 0.5 µM rotenone plus 0.5 mM Antimycin A (Sigma-Aldrich), with three consecutive determinations under each condition that were subsequently averaged. Nonmitochondrial respiration (OCR value after rotenone plus

antimycin A addition) was subtracted from all OCR measurements. ATP turnover was estimated from the difference between the basal and the oligomycin-inhibited respiration, and the maximal respiratory capacity was the rate in the presence of the uncoupler FCCP (*Brand and Nicholls, 2011*). Six independent replicas of each analysis were done, and results were normalized according to protein concentration.

## Multicolor fluorescence confocal microscopy

Lung samples were collected from autopsies performed on patients who died from diagnosed SARS-CoV-2 infection or unrelated causes (myocardial infarction, as controls) at the Gregorio Marañón Hospital, Madrid. Patient data and samples were obtained in accordance with the Ethics Committees of Instituto de Investigación Sanitaria Gregorio Marañón requirements. Formalin-fixed and paraffin-embedded tissues were deparaffinized, rehydrated, and unmasked by steaming in 10 mM sodium citrate buffer pH 9.0 (Dako Glostrup, Denmark) for 7 min. Slides were blocked with 5 µg/ml human immunoglobulins solved in blocking serum-free medium (Dako) for 30 min, and then sequentially incubated with 5–10 µg/ml primary antibodies, overnight at 4°C, and proper fluorescent secondary antibodies (Jackson Immunoresearch, West Grove, PA, USA) for 1 hr. Washes were performed by immersion in PBS containing 0.05% Tween-20, and single-cell quantification was performed in three to five fields imaged with the ACS_APO ×20/NA 0.60 objective of a confocal microscope (Leica, SPE). Mean fluorescence intensity of proteins of interest was obtained at segmented CD68+ macrophages using the 'analyze particle' plugin of ImageJ2 software (*Rueden et al., 2017*), as previously shown (*Gutiérrez-Seijo et al., 2022*).

## Statistical analysis

Statistical analyses were conducted using the GraphPad Prism software. For comparison of means, and unless otherwise indicated, statistical significance of the generated data was evaluated using the paired Student's t-test or one-way ANOVA followed by multiple comparisons using Dunnett's test or Fisher's LSD test. In all cases, $p < 0.05$ was considered statistically significant.

## Acknowledgements

This work was supported by grant PID2020-114323RB-I00 from Ministerio de Ciencia e Innovación to ALC, Dirección General de Innovación e Investigación Tecnológica de la Comunidad de Madrid (RETARACOVID, P2022/BMD-7274) to ALC, APK, PS-M, and RD, Instituto de Salud Carlos III (Grant PI23/00224 to APK, Grant PI21/00989 to RD), Red de Enfermedades Inflamatorias (RICORS RD21/0002/0034) from Instituto de Salud Carlos III and cofinanced by the European Regional Development Fund 'A way to achieve Europe' (ERDF) and PRTR to APK, and PID2021-123167OB-I00 from Ministerio de Ciencia, Innovación y Universidades and CSIC Talent Attraction program (20222AT010) to EO. This research work was also funded by the European Commission – NextGenerationEU (Regulation EU 2020/2094), through CSIC's Global Health Platform (PTI Salud Global). AN-V was funded by YEI program contract from Comunidad de Madrid (PEJ-2021-AI/BMD-23327). IR was funded by a Formación de Personal Investigador predoctoral fellowship from Ministerio de Ciencia e Innovación (Grant PRE2021-097080). Figures were created with Biorender.com.

## Additional information

### Competing interests

Israel Ríos: Cristina Herrero: The other author declares that no competing interests exist.

### Funding

| Funder | Grant reference number | Author |
| --- | --- | --- |
| Ministerio de Ciencia e Innovación | PID2020-114323RB-I00 | Angel L Corbí |

| Funder | Grant reference number | Author |
| --- | --- | --- |
| Comunidad de Madrid | RETARACOVID P2022/ BMD-7274 | Paloma Sánchez-Mateos Rafael Delgado Amaya Puig-Kröger Angel L Corbí |
| Instituto de Salud Carlos III | PI23/00224 | Amaya Puig-Kröger |
| Instituto de Salud Carlos III | PI21/00989 | Rafael Delgado |
| Instituto de Salud Carlos III | RICORS RD21/0002/0034 | Amaya Puig-Kröger |
| Ministerio de Ciencia, Innovación y Universidades | PID2021-123167OB-I00 | Eduardo Oliver |
| Ministerio de Ciencia e Innovación | FPI predoctoral fellowship PRE2021-097080 | Israel Ríos |
| Comunidad de Madrid | YEI program PEJ-2021-AI/ BMD-23327 | Alicia Nieto-Valle |
| European Commission – NextGenerationEU | EU 2020/2094 | Angel L Corbí |
| CSIC Talent Attraction program | 20222AT010 | Eduardo Oliver |

The funders had no role in study design, data collection and interpretation, or the decision to submit the work for publication.

## Author contributions

Israel Ríos, Conceptualization, Data curation, Formal analysis, Investigation, Methodology, Software, Supervision, Validation, Writing – original draft, Writing – review and editing; Cristina Herrero, Baltasar López-Navarro, Data curation, Formal analysis, Writing – review and editing; Mónica Torres-Torresano, Data curation, Formal analysis, Methodology, Project administration, Writing – review and editing; María Teresa Schiaffino, Francisco Díaz Crespo, Data curation, Formal analysis, Methodology, Validation, Visualization; Alicia Nieto-Valle, Data curation, Formal analysis, Visualization, Writing – review and editing; Rafael Samaniego, Data curation, Formal analysis, Methodology, Supervision, Visualization, Writing – review and editing; Yolanda Sierra-Palomares, Fernando Revuelta-Salgado, Ricardo García-Luján, Data curation, Formal analysis, Methodology, Writing – review and editing; Eduardo Oliver, Conceptualization, Data curation, Formal analysis, Funding acquisition, Investigation, Methodology, Supervision, Validation, Writing – review and editing; Paloma Sánchez-Mateos, Conceptualization, Data curation, Formal analysis, Funding acquisition, Investigation, Methodology, Supervision, Validation, Visualization, Writing – review and editing; Rafael Delgado, Conceptualization, Data curation, Formal analysis, Funding acquisition, Investigation, Supervision, Validation, Writing – review and editing; Amaya Puig-Kröger, Angel L Corbí, Conceptualization, Data curation, Formal analysis, Funding acquisition, Investigation, Methodology, Project administration, Resources, Supervision, Validation, Visualization, Writing – original draft, Writing – review and editing

## Author ORCIDs

Baltasar López-Navarro ⓘ https://orcid.org/0000-0002-4811-5419
Eduardo Oliver ⓘ https://orcid.org/0000-0001-9340-882X
Amaya Puig-Kröger ⓘ https://orcid.org/0000-0003-2943-9757
Angel L Corbí ⓘ http://orcid.org/0000-0003-1980-5733

## Ethics

Alveolar macrophages (AMØ) were obtained from remains of bronchoalveolar lavage (BAL) of patients undergoing bronchoscopy for diagnostic purposes under a protocol approved by the Internal Review Board of Instituto de Investigación Hospital 12 de Octubre (Reference TP24/0183). Lung samples were collected from autopsies performed on patients who died from diagnosed SARS-CoV-2 infection or unrelated causes (myocardial infarction, as controls) at the Gregorio Marañón Hospital, Madrid. Patient data and samples were obtained in accordance with Ethics Committees of Instituto de Investigación Sanitaria Gregorio Marañón requirements.

Reviewer #1 (Public review): https://doi.org/10.7554/eLife.102659.3.sa1
Author response https://doi.org/10.7554/eLife.102659.3.sa2

## Additional files

### Supplementary files
MDAR checklist

Supplementary file 1. Transcriptional datasets used in the present study.

Source code 1.

### Data availability
The dataset supporting the conclusions of this article is available in the Gene Expression Omnibus repository (http://www.ncbi.nlm.nih.gov/geo/) under accession GSE256538 (monocytes exposed to CHIR99021 or DMSO), GSE256208 (GM-MØ exposed to CHIR99021 or DMSO), GSE262463 (alveolar macrophages exposed to CHIR99021 or DMSO), and GSE266236 (GM-MØ after GSK3α/β knockdown). Computer code used for *Figure 7* is available in *Source code 1*.

The following datasets were generated:

| Author(s) | Year | Dataset title | Dataset URL | Database and Identifier |
|---|---|---|---|---|
| Ríos I, Herrero C, Puig-Kröger A, Corbí AL | 2025 | Effect of GSK3 inhibition on the transcriptional signature of human monocytes | https://www.ncbi.nlm.nih.gov/geo/query/acc.cgi?acc=GSE256538 | NCBI Gene Expression Omnibus, GSE256538 |
| Ríos I, Torres-Torresano M, Puig-Kröger A, Corbí AL | 2025 | Effect of GSK3 inhibition on the transcriptional signature of human GM-CSF-dependent monocyte-derived macrophages | https://www.ncbi.nlm.nih.gov/geo/query/acc.cgi?acc=GSE256208 | NCBI Gene Expression Omnibus, GSE256208 |
| Ríos I, Delgado R, Puig-Kröger A, Corbí AL | 2025 | Effect of GSK3 inhibition on the transcriptional signature of human alveolar macrophages | https://www.ncbi.nlm.nih.gov/geo/query/acc.cgi?acc=GSE262463 | NCBI Gene Expression Omnibus, GSE262463 |
| Ríos I, Puig-Kröger A, Corbí AL | 2025 | Effect of siRNA-mediated GSK3 knock-down on the transcriptional signature of human GM-CSF-dependent monocyte-derived macrophages | https://www.ncbi.nlm.nih.gov/geo/query/acc.cgi?acc=GSE266236 | NCBI Gene Expression Omnibus, GSE266236 |

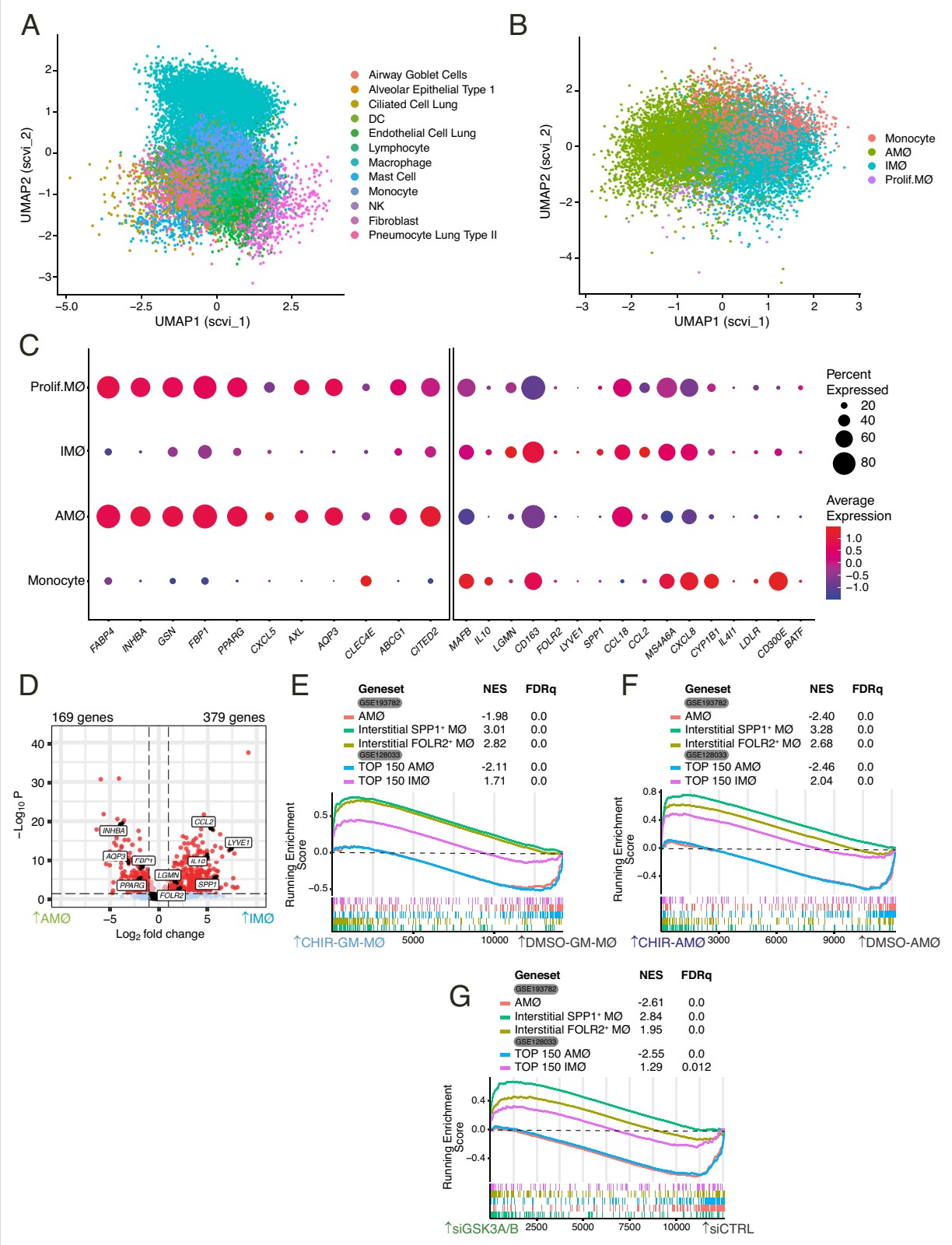

**Figure 7.** Differential expression of GSK3-regulated genes in human lung interstitial and alveolar macrophages. (**A**) Embedding of 21,310 single-cell transcriptomes from human lungs in UMAP (Uniform Manifold Approximation and Projection) with the scVI reduction. Cell-type annotation is based on the expression of canonical marker genes, as reported in *Morse et al., 2019*. (**B**) Two-dimensional embedding computed by UMAP with the scVI reduction on computationally identified macrophages after filtering according to number of genes per cell (nFeature, >200 and <6000), Unique

*Figure 7 continued on next page*

*Figure 7 continued*

Molecular Identifiers (nCount, >1000), and % of mitochondrial genes (<15%). (**C**) Relative expression of the indicated genes in the four macrophage subsets defined upon re-clustering of the macrophages identified in the single-cell RNA sequencing reported in GSE128033 (*Morse et al., 2019*). (**D**) Volcano plot illustrating the differentially expressed genes between the interstitial macrophage (IMØ) and alveolar macrophage (AMØ) subsets defined upon re-analysis of the single-cell RNA sequencing reported in GSE128033 (*Morse et al., 2019*). (**E–G**) Gene set enrichment analysis (GSEA) of the gene sets that define human lung AMØ or IMØ macrophage subsets (GSE128033 and GSE193782) (*Morse et al., 2019*; *Li, 2022*) on the ranked comparison of CHIR-GM-MØ vs. DMSO-GM-MØ transcriptomes (**E**), CHIR-AMØ vs. DMSO-AMØ transcriptomes (**F**), or siGSK3A/B-GM-MØ vs. siCNT-GM-MØ (**G**).

The online version of this article includes the following figure supplement(s) for figure 7:

**Figure supplement 1.** Number of cells from single-cell RNA sequencing on human lungs from eight independent donors (from GSE128033) (*Morse et al., 2019*), after filtering according to number of genes per cell (nFeature, >200 and <6000), Unique Molecular Identifiers (nCount, >1000) and % of mitochondrial genes (<15%).

**Figure supplement 2.** UMAP (Uniform Manifold Approximation and Projection), with the scVI reduction, embedding of 21,310 single-cell transcriptomes from human lungs as reported in GSE128033 (*Morse et al., 2019*), with color-coding indicating distinct cell clusters.

**Figure supplement 3.** UMAP (Uniform Manifold Approximation and Projection), with the scVI reduction, embedding of 21,310 single-cell transcriptomes from human lungs as reported in GSE128033 (*Morse et al., 2019*), with color-coding indicating donor origin.

**Figure supplement 4.** Color-coded expression of the indicated genes after projection onto the UMAP (Uniform Manifold Approximation and Projection) embedding shown in *Figure 7—figure supplements 2 and 3*.

**Figure supplement 5.** UMAP (Uniform Manifold Approximation and Projection), with the scVI reduction, embedding of single-cell transcriptomes from human lung macrophages as reported in GSE128033 (*Morse et al., 2019*), with color-coding indicating distinct cell clusters.

**Figure supplement 6.** UMAP (Uniform Manifold Approximation and Projection), with the scVI reduction, embedding of single-cell transcriptomes from human lungs as reported in GSE128033 (*Morse et al., 2019*), with color-coding indicating donor origin.

**Figure supplement 7.** Color-coded expression of the genes used for macrophage definition (*CD163*, *FABP4*, *LYVE1*, *FCN1*), as well as proliferation-associated genes (*TYMS*, *MKI67*, *TOP2A*, *NUSAP1*) and other bona fide macrophage marker genes (*SPI1*, *FOLR2*), after projection onto the UMAP embedding as shown in *Figure 7—figure supplements 5 and 6*.

The following previously published datasets were used:

| Author(s) | Year | Dataset title | Dataset URL | Database and Identifier |
|---|---|---|---|---|
| Grant RA, Morales-Nebreda L, Markov NS, Swaminathan S, Querrey M, Guzman ER, Abbott DA, Donnelly HK, Donayre A, Goldberg IA, Klug ZM, Borkowski N, Lu Z, Kihshen H, Politanska Y, Sichizya L, Kang M, Shilatifard A, Qi C, Lomasney JW, Argento AC, Kruser JM, Malsin ES, Pickens CO, Smith SB, Walter JM, Pawlowski AE, Schneider D, Nannapaneni P, Abdala-Valencia H, Bharat A, Gottardi CJ, Budinger GS, Misharin AV, Singer BD, Wunderink RG | 2020 | Circuits between infected macrophages and T cells in SARS-CoV-2 pneumonia | https://www.ncbi.nlm.nih.gov/geo/query/acc.cgi?acc=GSE155249 | NCBI Gene Expression Omnibus, GSE155249 |
| Liao M, Liu Y, Yuan J, Wen Y, Xu G, Zhao J, Cheng L, Li J, Wang X, Wang F, Liu L, Amit I, Zhang S, Zhang Z | 2020 | Single-cell landscape of bronchoalveolar immune cells in COVID-19 patients | https://www.ncbi.nlm.nih.gov/geo/query/acc.cgi?acc=GSE145926 | NCBI Gene Expression Omnibus, GSE145926 |
| Muller IB, Jansen G, Kröger AP | 2022 | Gene profile of human CD14+ monocytes and monocyte-derived macrophages | https://www.ncbi.nlm.nih.gov/geo/query/acc.cgi?acc=GSE188278 | NCBI Gene Expression Omnibus, GSE188278 |
| Dominguez-Soto A, Vega MA, Corbí AL | 2020 | Transcriptional effects of MAF- and MAFB-siRNAs on M-CSF-derived macrophages | https://www.ncbi.nlm.nih.gov/geo/query/acc.cgi?acc=GSE155719 | NCBI Gene Expression Omnibus, GSE155719 |
| Vega MA, Simón-Fuentes M, Domínguez-Soto A, Corbí AL | 2023 | MAFB-binding sites by ChIP-Seq on human M-CSF derived macrophages | https://www.ncbi.nlm.nih.gov/geo/query/acc.cgi?acc=GSE190589 | NCBI Gene Expression Omnibus, GSE190589 |
| Li X, Kolling FW, Aridgides D, Mellinger D, Ashare A, Jakubzick CV | 2022 | ScRNA-seq Expression of APOC2 and IFI27 Identifies Four Alveolar Macrophage Superclusters in Cystic Fibrosis and Healthy BALF | https://www.ncbi.nlm.nih.gov/geo/query/acc.cgi?acc=GSE193782 | NCBI Gene Expression Omnibus, GSE193782 |
| Morse C, Tabib T, Sembrat J, Trejo Bittar HE, Buschur K, Valenzi E, Jiang Y, Kass D, Gibson K, Chen W, Mora A, Benos P, Rojas M, Lafyatis R | 2019 | Proliferating SPP1/MERTK-expressing macrophages in idiopathic pulmonary fibrosis | https://www.ncbi.nlm.nih.gov/geo/query/acc.cgi?acc=GSE128033 | NCBI Gene Expression Omnibus, GSE128033 |
| Wendisch D, Dietrich O, Mari T | 2021 | Single-cell RNA-seq of bronchoalveolar lavage (BAL) fluid of late stage severe COVID-19 patients | https://ega-archive.org/studies/EGAS00001005634 | European Genome-Phenome Archive (EGA), EGAS00001005634 |

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
