## [Editor Report · eLife Assessment]

This **important** study provides **compelling** data from in vitro models and patient-derived samples to demonstrate how modulation of GSK3 activity can reprogram macrophages, revealing potential therapeutic applications in inflammatory diseases such as severe COVID-19. The study stands out for its clear and systematic presentation, **convincing** experimental approach, and the relevance of its findings to the field of immunology.

---

## [Referee Report · Reviewer #1 (Public review)]

The manuscript by Rios et al. investigates the potential of GSK3 inhibition to reprogram human macrophages, exploring its therapeutic implications in conditions like severe COVID-19. The authors present convincing evidence that GSK3 inhibition shifts macrophage phenotypes from pro-inflammatory to anti-inflammatory states, thus highlighting the GSK3-MAFB axis as a potential therapeutic target. Using both GM-CSF- and M-CSF-dependent monocyte-derived macrophages as model systems, the study provides extensive transcriptional, phenotypic, and functional characterizations of these reprogrammed cells. The authors further extend their findings to human alveolar macrophages derived from patient samples, demonstrating the clinical relevance of GSK3 inhibition in macrophage biology.

The experimental design is sound, leveraging techniques such as RNA-seq, flow cytometry, and bioenergetic profiling to generate a comprehensive dataset. The study's integration of multiple model systems and human samples strengthens its impact and relevance. The findings not only offer insights into macrophage plasticity but also propose novel therapeutic strategies for macrophage reprogramming in inflammatory diseases.

Strengths:

(1) Robust Experimental Design: The use of both in vitro and ex vivo models adds depth to the findings, making the conclusions applicable to both experimental and clinical settings.

(2) Thorough Data Analysis: The extensive use of RNA-seq and gene set enrichment analysis (GSEA) provides a clear transcriptional signature of the reprogrammed macrophages.

(3) Relevance to Severe COVID-19: The study's focus on macrophage reprogramming in the context of severe COVID-19 adds clinical significance, especially given the relevance of macrophage-driven inflammation in this disease.

Weaknesses:

There are no significant weaknesses in the study.

---

## [Author Response]

The following is the authors’ response to the original reviews

**Reviewer #1 (Public review):**
The manuscript by Rios et al. investigates the potential of GSK3 inhibition to reprogram human macrophages, exploring its therapeutic implications in conditions like severe COVID-19. The authors present convincing evidence that GSK3 inhibition shifts macrophage phenotypes from pro-inflammatory to anti-inflammatory states, thus highlighting the GSK3-MAFB axis as a potential therapeutic target. Using both GM-CSF- and M-CSF-dependent monocyte-derived macrophages as model systems, the study provides extensive transcriptional, phenotypic, and functional characterizations of these reprogrammed cells. The authors further extend their findings to human alveolar macrophages derived from patient samples, demonstrating the clinical relevance of GSK3 inhibition in macrophage biology.The experimental design is sound, leveraging techniques such as RNA-seq, flow cytometry, and bioenergetic profiling to generate a comprehensive dataset. The study's integration of multiple model systems and human samples strengthens its impact and relevance. The findings not only offer insights into macrophage plasticity but also propose novel therapeutic strategies for macrophage reprogramming in inflammatory diseases.Strengths:(1) Robust Experimental Design: The use of both in vitro and ex vivo models adds depth to the findings, making the conclusions applicable to both experimental and clinical settings.(2) Thorough Data Analysis: The extensive use of RNA-seq and gene set enrichment analysis (GSEA) provides a clear transcriptional signature of the reprogrammed macrophages.(3) Relevance to Severe COVID-19: The study's focus on macrophage reprogramming in the context of severe COVID-19 adds clinical significance, especially given the relevance of macrophage-driven inflammation in this disease.Weaknesses:There are no significant weaknesses in the study, though some minor points could be addressed for clarity and completeness, as outlined in the recommendations below.

Many thanks for these comments. Please find below the response to the specific recommendations.

**Recommendations for the authors**:(1) In lines 263-266, the term "MoMac-VERSE" and its associated clusters are introduced without sufficient explanation. The authors should provide additional clarification on what these clusters represent and how they were derived.

We have revised the text according to the reviewer´s suggestion and followed the original nomenclature of the MoMac-VERSE monocyte/macrophage clusters, also recognizing the procedure for their identification. The newly modified text now states: "Thus, analysis of the MoMac-VERSE (a resource that identified conserved monocyte and macrophage states derived from healthy and pathologic human tissues) (GSE178209) (2), indicated that GSK3 inhibition augments the expression of the gene sets that define MoMac-VERSE subsets identified as long-term resident macrophages [Cluster HES1_Mac (#2)] and tumor-associated macrophages with an M2-like signature [Clusters HES1_Mac (#2), TREM2_Mac (#3), C1Q^hi^_Mac (#16) and FTL_Mac (#17)] (2) (Figure 1H)."

(2) In line 283, the reference labeled "2227" appears incorrect. It seems to be a formatting issue, and it might refer to references 22-27. Please verify and correct.

All wrongly formatted references throughout the manuscript have been checked and corrected.

(3) In line 353, the reference is incorrect. Please reviewe ensure that all references are properly cited throughout the manuscript.

All wrongly formatted references throughout the manuscript have been checked and corrected.

(4) In line 368, one of the patient samples shows a decreased IL-10 response after CHIR treatment. The authors should acknowledge the heterogeneity in the primary cell responses and adjust the conclusion accordingly to reflect this variability.

We have modified the text following the reviewer´s comment, and acknowledge the heterogeneity in the production of IL-10 after GSK3 inhibition in the three analyzed samples. The modified text now states: "Consistent with these findings, CHIR-AMØ exhibited higher expression of MAFB (Figure 6F) whose increase correlated with an augmented secretion of Legumain, CCL2 and IL-10 (Figure 6G), although the latter was only seen in two samples, probably reflecting heterogeneity in primary cell responses."

(5) Figure 7B: the UMAP shows 4 populations, but according to the visualization in the sup fig 3, there should be many more clusters. How do the authors explain this? Are these patient-specific clusters? Also, IMs can be separated into at least subpopulations. Can the authors plot also bona fide macrophage markers expressed by all subpopulations?

To clarify this whole issue, and avoid misleading visualization of donor-specific clusters (see below), we have now replaced all UMAP plots shown in the previous version (in old Figure 7 and old Supplementary Figure 3) with new UMAP plots after running scVI reduction. In addition, we are including a new Supplementary Figure (new Supplementary Figure 3) that contains the information of the 21310 single-cell transcriptomes from human lungs reported in GSE128033 (ref. 47) after filtering and integration [nFeature > 200 and < 6000; Unique Molecular Identifiers (nCount > 1000) and % of mitochondrial genes (< 15 %)]. Besides, old Supplementary Figure 3 has been replaced by the new Supplementary Figure 4, which includes the information of the single-cell transcriptomes from human lung macrophages selected from GSE128033 (ref. 47) based on their expression of the monocyte/macrophage-associated markers *CD163*, *FABP4*, *LYVE1* or *FCN1*.

Addresing the first question, UMAPs in old Figure 7B and old Supplementary Figure 3B had a different number of clusters because old Figure 7B was derived from old Supplementary Figure 3B after grouping macrophage clusters according to the expression of previously defined markers and to limit the weight of donor-specific clusters. Specifically, the macrophage clusters from old Figure 7B were re-grouped according to the differential expression of:

*- FCN1* (including cluster 4, 7 and 12 from Figure 7B): Infiltrating monocytes.

- *FABP4* and *TYMS*-negative (including clusters 0, 2, 5 and 13 from Figure 7B), or MARCO and INHBA (cluster 9 from Figure 7B) or PPARG (cluster 11 from Figure 7B): Alveolar macrophages (AMØ).

- *TYMS*, *MKI67*, *TOP2A* and *NUSAP1* (cluster 15 from Figure 7B): Proliferating AMØ.

- *LYVE1* or *RNASE1* or *LGMN* (including clusters 1, 3, 6, 8, 10 and 14 from Figure 7B): Interstitial Macrophages (IMØ).

As the reviewer suggested, this type of UMAP plot yielded a large number of donor-specific clusters. To avoid such a misleading representation, we have now plotted UMAPs after running scVI reduction in every case. The new plots are now shown in new Figure 7A, new Figure 7B, new Supplementary Figure 3 (containing the information of the 21310 single-cell transcriptomes from GSE128033) and the novel Supplementary Figure 4 (with the information of the single-cell transcriptomes from human lung macrophages from GSE128033).

Finally, to address the last issue, we have now plotted the expression of genes used for macrophage definition (*CD163, FABP4, LYVE1, FCN1*), as well as proliferation-associated genes (*TYMS, MKI67, TOP2A, NUSAP1*) and other *bona fide* macrophage marker genes (*SPI1, FOLR2*) in Supplementary Figure 4C.

(6) statistics should be indicated in every figure legend and for every subfigure where applicable.

We have now included the specific statistical procedure applied for each Figure and panel.

**Reviewer 2 (Public review):**
The study by Rios and colleagues provides the scientific community with a compelling exploration of macrophage plasticity and its potential as a therapeutic target. By focusing on the GSK3-MAFB axis, the authors present a strong case for macrophage reprogramming as a strategy to combat inflammatory and fibrotic diseases, including severe COVID-19. Using a robust and comprehensive methodology, in this study it is conducted a broad transcriptomic and functional analyses and offers valuable mechanistic insights while highlighting its clinical relevanceStrengths:Well performed and analyzedWeaknesses:Additional analyses, including mechanistic studies, would increase the value of the study

In an effort to address the comment of the reviewer, we have performed more detailed analysis of the kinetics and dose-response effects of GSK3 inhibition, which are now provided as new Supplementary Figure 3A.

Regarding additional mechanistic studies, we decided to explore the relationship between inactive GSK3β and MAFB levels at the early stages of M-CSF- or GM-CSF-driven monocyte-to-macrophage differentiation. These experiments, performed in three independent monocyte preparations, indicated that, 48 hours along differentiation, M-CSF promoted a huge increase in both MAFB expression and a slight (albeit significant) rise in inactive GSK3β (P-Ser9-GSK3β) (compared to either untreated or GM-CSF-treated monocytes), further supporting the macrophage re-programming effect of GSK3. However, since the M-CSF-promoted increase in MAFB levels was much robust than the enhancement in inactive GSK3β, we hypothesize that proteasomal degradation of MAFB might be also distinct between M-CSF- (M-MØ) and GM-CSF-dependent (GM-MØ) monocyte-derived macrophages.

**Author response image 1. sa2fig1:** Total GSK3β, p-Ser9-GSK3β and MAFB levels in three preparations of freshly purified monocytes either unstimulated (-) or stimulated with M-CSF (10 ng/ml) or GM-CSF (1,000 U/ml) at different time points, as determined by Western blot (upper panel). Vinculin protein levels were determined as protein loading control. Mean ± SEM of the GSK3β/Vinculin, p-Ser9-GSK3β/Vinculin, and MAFB/Vinculin protein ratios from the three independent experiments are shown (lower panel) (paired Student’s t test: *, p<0.05; ****, p<0.001).

Based on this finding, we then determined proteasome activity in fully differentiated M-CSF- and GM-CSF-dependent monocyte-derived macrophages. Use of the Immunoproteasome Activity Fluorometric Assay Kit II (UBPBio) in M-MØ and GM-MØ, either untreated or exposed to the proteasome inhibitor MG132, revealed that immune-proteasomal and proteasomal activity is significantly stronger in GM-MØ than in M-MØ, as demonstrated in assays for chymotrypsin-like (ANW) and branched amino acid preferring (PAL) activity (immunoproteasome), and trypsin-like (KQL) activity (both proteasome and immunoproteasome). This result suggested that, indeed, immunoproteasomal activity might contribute to the differential expression of MAFB in M-MØ and GM-MØ.

**Author response image 2. sa2fig2:** Immunoproteasome activity in M-MØ and GM-MØ, either untreated or exposed to MG132, as determined using the Immunoproteasome Activity Fluorometric Assay Kit II (UBPBio) on the three indicated peptides (upper panel). Mean ± SEM of three independent experiments are shown (paired Student’s t test: *, p<0.05) (lower panel).

Consequently, we next set up experiments to assess whether the proteasome inhibitor MG132 was capable of enhancing the expression of MAFB-dependent genes in GM-MØ. Preliminary results of GM-MØ exposure to MG132 for 6 hours indicated an increase in the expression of MAFB protein and the MAFB-dependent genes *LGMN* and *IL10*. , as well as a reduction in the expression of the GM-MØ-specific gene *CD1C*.

**Author response image 3. sa2fig3:** A. Schematic representation of the exposure of MG132 to GM-MØ for 6 hours. B. MAFB protein levels in four independent preparations of GM-MØ exposed to either DMSO (DMSO-GM-MØ) or the proteasome inhibitor MG132 (MG132-GM-MØ) for 6 hours, as determined by Western blot (left panel). GAPDH protein levels were determined as protein loading control. Mean ± SEM of the MAFB/GAPDH protein ratios from the four independent experiments are shown (right panel) (paired Student’s t test: ***, p<0.005). C. Relative mRNA levels of the indicated genes in DMSO-GM-MØ and MG132-GM-MØ, as determined by RT-PCR on seven independent samples (paired Student’s t test: ***, p<0.005; ****, p<0.001).

Unfortunately, this proteasome inhibitor (MG-132) caused a great reduction in cell viability after 6-8 hours. Since a similar decrease in cell viability was observed upon analysis with the ONX-0914 immunoproteasome inhibitor, we could not procede any further with this approach.

Given the reviewer´s suggestion to include mechanistic insights to the manuscript, we are now providing these results (and the corresponding figures) only for the reviewer´s information and to make clear our attempts to comply with his/her request.

**Recommendations for the authors**:The results are of interest, and only some minor issues need to be addressed to strengthen the conclusions of the study.

We gratefully thank the reviewer for his/her comments.

(1) This study employs a single dose of 10 μM of the GSK3 inhibitor CHIR-99021 for 48 hours, which is reasonable for in vitro studies. However, further investigation into the effect of different doses and exposure times could provide additional insight into optimal dosing and durability of reprogramming effects. In addition, would an alternative GSK3 inhibitors have comparable effects?

Following the reviewer suggestion, we have performed a kinetics and dose-response analysis of the effects of CHIR-99021, using MAFB protein levels as a readout. This experiments is now shown in new Supplementary Figure 1A, that replaces the old Supplementary Figure 1A panel where a shorter kinetics was presented. Results of this new experiment indicates a maximal effect of 10µM CHIR-99021, and that the effect of the inhibitor becomes maximal 24-48 hours after treatment. The text has been modified accordingly, and it now states: "Kinetics and dose-response analysis of the effects of CHIR-99021 on MAFB expression showed that maximal protein levels were achieved after a 24-48 hour exposure to 10µM CHIR-99021 (Supplementary Figure 1A), conditions that were used hereafter."

Regarding the use of alternative GSK3 inhibitors, we had already provided that information in Supplementary Figure 1B, where the effects of SB-216763 (10 µM) or LiCl (10 mM) were evaluated. The huge reversal of the Tyr^216^/Ser^9^ GSK3β phosphorylation ratio observed with CHIR-99021 was not seen with other GSK3 inhibitors, as indicated in the text. In any event, we believe that the relevance of this result with SB-216763 or LiCl is minimized by the results generated after siRNA-mediated GSK3 knockdown (shown in Figure 4), that completely reproduced the effects seen with CHIR-99021.

(2) Why in the "reanalysis of single cell RNAseq data" section, the authors use Seurat v5 (R) but then change to python, and the other way around?

As indicated in the documentation for Integrative Analysis in Seurat v5 (https://satijalab.org/seurat/articles/seurat5_integration), scVIIntegration requires reticulate package which allow us to run Python environment in R.

(3) When the authors refer to the clusters enriched in MoMacVERSE, they use the labels of the clusters (for example #2 or #3). I would suggest using the annotations described in the original paper, to link it to the bibliography published through the labels established in the paper.

We have revised the text according to the reviewer´s suggestion and followed the original nomenclature of the MoMac-VERSE monocyte/macrophage clusters, also recognizing the procedure for their identification. The newly modified text now states: "Thus, analysis of the MoMac-VERSE (a resource that identified conserved monocyte and macrophage states derived from healthy and pathologic human tissues) (GSE178209) (2), indicated that GSK3 inhibition augments the expression of the gene sets that define MoMac-VERSE subsets identified as long-term resident macrophages [Cluster HES1_Mac (#2)] and tumor-associated macrophages with an M2-like signature [Clusters HES1_Mac (#2), TREM2_Mac (#3), C1Q^hi^_Mac (#16) and FTL_Mac (#17)] (2) (Figure 1H)."

(4) In line 309. Is there any significance on the "having a stronger effect"?

We apologize for the misleading sentence. The phrase has been modified for better clarity, and the text now states: "Like CHIR-99021, silencing of both GSK3A and GSK3B augmented the expression of MAFB, with the simultaneous silencing of both GSK3A and GSK3B genes having a stronger effect (Figure 4B), and modulated the expression of 329 genes (Figure 4C,D)."

(5) In line 337, "(22)(27)", are these references?

All wrongly formatted references throughout the manuscript have been checked and corrected.

(6) In the single-cell reanalysis, could you please provide integration Qc plots? It would be interesting to have it on the paper.

To clarify this whole issue, and avoid misleading visualization of donor-specific clusters (see below), we have now replaced all UMAP plots shown in the previous version (in old Figure 7 and old Supplementary Figure 3) with new UMAP plots after running scVI reduction. In addition, we are including a new Supplementary Figure (new Supplementary Figure 3) that contains the information of the 21310 single-cell transcriptomes from human lungs reported in GSE128033 (ref. 47) after filtering and integration [nFeature > 200 and < 6000; Unique Molecular Identifiers (nCount > 1000) and % of mitochondrial genes (< 15 %)]. Besides, old Supplementary Figure 3 has been replaced by the new Supplementary Figure 4, which includes the information of the single-cell transcriptomes from human lung macrophages selected from GSE128033 (ref. 47) based on their expression of the monocyte/macrophage-associated markers *CD163*, *FABP4*, *LYVE1* or *FCN1*.

As requested by the reviewer, we are now providing the Qc plots for the re-analysis in the new Supplementary Figures 3 and 4.